# VFedCD: Vertical Federated Learning-based Causal Discovery

## Abstract

Causal discovery seeks to identify causal relationships among attributes, typically represented as directed acyclic graphs (DAGs) where vertices denote attributes and edges denote direct causal effects. Existing methods struggle in vertically federated scenarios. In these settings, data is partitioned across parties that hold disjoint attributes, and strict privacy constraints prevent centralized aggregation, leaving vertical federated causal discovery underexplored. We propose VFedCD, the first framework for causal discovery in vertical federated settings. VFedCD models causal mechanisms with a shallow-encoder, deep-decoder design. Each party uses a shallow encoder to transform its local attributes into privacy-preserving features for all parties, and then a deep decoder to aggregate received features and predict local attributes, implicitly capturing causal dependencies. To avoid cycles or overly dense graph structures, a Centralized Topology Validator (CTV) extracts partial causal structures from party encoders, aggregates them into a global graph and enforces structural constraints. In addition, a Secure Dispatch Protocol (SDP) is designed to enhance the security of feature exchange and gradient propagation by redesigning encoding and aggregation with semi-homomorphic encryption and secret sharing. Experiments on synthetic and real-world datasets with artificial vertical partitioning show that VFedCD matches the accuracy of centralized methods while guaranteeing privacy.

## 1 Introduction

Causal discovery, which aims to uncover directed cause-effect relationships among attributes, has become a cornerstone of scientific inquiry in diverse domains, including biomedical research (Imbens & Rubin, 2015), climate science (Zhang et al., 2011), and epidemiology (Greenland et al., 1999). Traditional causal discovery methods typically assume centralized access to complete data. Among them, a popular branch is *Differentiable Causal Discovery* (DCD) (Zheng et al., 2018). DCD leverages neural networks to model causal mechanisms by taking all attributes as input to predict a target attribute, approximating causal relationships through model fitting, and deriving causal graphs from the learned parameters.

However, in many real-world scenarios, attributes of the same samples are often vertically partitioned across multiple parties, with each party holding only a subset of attributes (Yang et al., 2019; Liu et al., 2020b). As illustrated in Fig. 1(a), consider a medical study aiming to infer causal relationships among basic health information, genetic data, and exercise habits of citizens, while these data are stored separately in hospitals, medical examination institutions, and fitness management Apps. Due to privacy regulations and commercial constraints, raw data sharing is prohibited, rendering centralized approaches to causal discovery infeasible.

To address this challenge, we propose VFedCD, a framework based on Vertical Federated Learning (VFL) for DCD. In VFedCD, parties collaboratively infer causal relationships without sharing raw data. VFL is a classic federated learning paradigm that enables joint model training over vertically partitioned data by transmitting encoded features and gradients between parties instead of raw data (Liu et al., 2020a; Hu et al., 2019; Wan et al., 2007). For instance, to evaluate the causal effect of Institution A's attributes on Hospital C's (Fig. 1(b)), Institution A encodes its local attributes into features and sends them to C. Hospital C then uses these features to predict its own attributes, and the resulting gradients are sent back to A to update its encoder. Through this iterative process of

exchanging privacy-preserving intermediate representations, the causal graph can be inferred from the optimized model parameters.

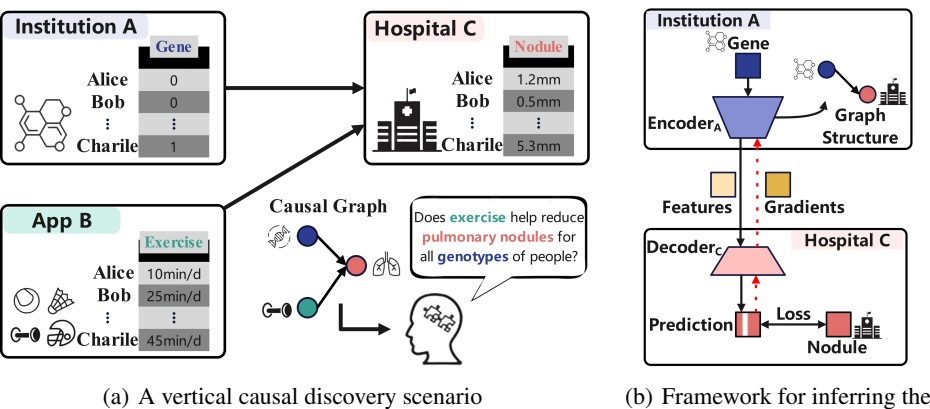

(a) A vertical causal discovery scenario

(b) Framework for inferring the causal effect of A on C

Figure 1: Vertical Federated Causal Discovery: Scenario and Basic Framework

Unlike traditional VFL frameworks focus on discriminative tasks (e.g., classification), applying it directly to causal discovery presents unique challenges. Decisions on how to split the model structure inevitably involve a trade-off between performance and privacy. Transmitting shallow-layer features typically yields better performance but also increases the risk of privacy leakage due to their higher fidelity to raw data.

To address this, we adopt a shallow-encoder deep-decoder (sEdD) architecture to enhance inter-party causal mechanism modeling. To mitigate privacy risks from linear features generated by shallow encoders, we design a Secure Dispatch Protocol (SDP) that combines semi-homomorphic encryption and secret sharing to decompose feature computations into encrypted fragments, ensuring no party has access to standalone features or gradients during training (Fu et al., 2022). Additionally, differentiable causal discovery requires enforcing global acyclicity constraints on the causal graph. For this, a Centralized Topology Validator (CTV) is introduced to aggregate causal graph structures from local encoder parameters and enforce global acyclicity constraints. This effectively prevents cyclic or overly dense structures while avoiding privacy risks brought by sensitive graph structures exchange among parties.

Extensive experiments demonstrate VFedCD's efficacy across synthetic and real-world datasets. VFedCD achieves causal discovery accuracy comparable to centralized methods while providing provable privacy guarantees. Our work makes three key contributions:

1. We establish a theoretical foundation for causal discovery in vertically federated learning, identifying unique challenges and opportunities in distributed attribute settings.

2. We propose VFedCD, which, to the best of our knowledge, is the first vertical federated causal discovery framework. It introduces two key components—the CTV and the SDP—to enable global constraint enforcement and privacy-preserving feature aggregation, respectively.

3. Extensive experiments on both synthetic and real-world datasets demonstrate that VFedCD achieves performance comparable to centralized causal discovery methods while providing privacy guarantees.

## 2 RELATED WORK

**Causal discovery** methods primarily target centralized data. Constraint-based (e.g., PC (Kalisch & Bühlman, 2007), BPC (Harris & Drton, 2013), IDA (Maathuis et al., 2009)), score-based (e.g., Greedy Equivalence Search (GES) (Chickering, 2002), CD2 (Gu et al., 2019), SP (Raskutti & Uhler, 2018)), and mechanism-fitting (e.g., NO-TEARS (Zheng et al., 2018), LiNGAM (Shimizu et al.,

2006), GES (Chickering, 2002), SDCD (Nazaret et al., 2023)) approaches all assume access to full attribute sets, which is infeasible for vertical federation where attributes are disjoint across parties.

**Federated causal discovery (FCD)** has focused on horizontal federation, where parties share the same attributes but different samples. These methods fall into two categories: 1) *Party-driven* approaches (e.g., FedDAG (Gao et al., 2021), FedCSL (Guo et al., 2024a), Bloom (Chengbo & Kai, 2024), NOTEARS-ADMMTh (Ng & Zhang, 2022), FedACD (Guo et al., 2024b), FedGES (Torrijos et al., 2024)) enable local causal graph construction or neighborhood learning, with servers aggregating parameters or selecting optimal structures; 2) *Server-led* approaches (e.g., FedC2SL (Wang et al., 2023), PERI (Mian et al., 2023), LiNGAMs (Shimizu, 2012), DARLS (Ye et al., 2024), FedCDH (Li et al., 2024)) use parties to validate global hypotheses (e.g., topological order, graph or independence test) for centralized graph refinement. Both rely on parties having complete attribute sets, a critical assumption invalid in vertical federation—no party holds all attributes, making hypothesis validation and causal mechanism modeling across disjoint attributes impossible.

## 3 PRELIMINARIES

**Assumption of causal discovery.** Our approach is based on two assumptions. The first assumption is the Additive Noise Model (ANM), where each attribute $X_j$ is generated from its parent attributes $PA_j$ through causal mechanism $f_j$ with independent additive noise $\epsilon_j$: (Peters & Bühlmann, 2014): $X_j = f_j(\mathbf{PA}_j) + \epsilon_j$, $\epsilon_j$ representing independently distributed noise. The second is the Faithfulness assumption, which states that the conditional independencies in the data align with the $d$-separation properties underlying the DAG (Spirtes et al., 2001). Discussion on the impact of these assumptions on identifiability is provided in the appendix E.

**Homomorphic encryption (HE).** HE defines an encryption function $E$ and decryption function $D$. For a plaintext message $m$ and public key $pub$, encryption is given by $\|m\|_{pub} = E(m, pub)$, and decryption with the private key $pri$ is $m = D(\|m\|_{pub}, pri)$. Additive HE (e.g., the Paillier scheme (Paillier, 2005)) supports homomorphic addition $\|m_1\| \oplus \|m_2\| = \|m_1 + m_2\|$ and scalar multiplication $c \otimes \|m_1\| = \|c \cdot m_1\|$, enabling computations on encrypted data without the need for decryption.

**Secret sharing of homomorphic encryption ciphertexts (HE2SS).** HE2SS converts additive HE ciphertexts to Secret Shares (Du & Atallah, 2001), as illustrated in Algorithm 1. Given that party $p_A$ holds $\|data\|_B = E(data, p_B.pub)$ (i.e., data encrypted with party $p_B$'s public key via homomorphic encryption (Paillier, 2005)), the procedure is as follows: 1. Party $p_A$ generates random noise $\epsilon$ and computes $\|data - \epsilon\|_B = \|data\|_B \oplus E(-\epsilon, p_B.pub)$. 2. Party $p_A$ sends $\|data - \epsilon\|_B$ to party $p_B$, which decrypts it using $p_B.pri$ to obtain $data - \epsilon$. 3. The result is secret-shared as $[\epsilon, data - \epsilon]_{A,B}$, where $p_A$ holds $\epsilon$ and $p_B$ holds $data - \epsilon$, ensuring that neither party can reconstruct $data$ alone.

## 4 PROBLEM DEFINITION

**Vertically Partitioned Data.** Consider $K$ parties $\{p_k\}$ holding $N$ samples with vertically partitioned features, where party $k$ holds local data $D_k = \{x_n^k\}$. Each sample $x_n \in \mathbb{R}^d$ is split into $x_n^k \in \mathbb{R}^{d_k}$ for party $k$, with $\sum d_k = d$ (Yang et al., 2019).

**Learning Objective.** The goal is to collaboratively learn a causal graph $B = \text{graph}(\Theta)$ from vertically partitioned data $D = \{D_k\}$, where the parameter set $\Theta$ includes all encoders $\{\phi_{kt}\}$ and decoders $\{\omega_k\}$ between parties, respectively responsible for encoding the attributes of party k into required features for target party t and decoding the aggregated features to predict the attributes of party k. To facilitate notation, we define the parameters of each party as $\theta_k = \{\phi_{kt}\}_{t=1}^K \cup \{\omega_k\}$, so that the complete set of parameters is $\Theta = \{\theta_k\}_{k=1}^K$. The joint objective minimizes the prediction loss $l$, regularizes model complexity via a penalty term $\alpha$ with hyperparameter $\lambda_1$, and enforces a continuous acyclicity constraint $h(B)$ (Nazaret et al., 2023) with hyperparameter $\lambda_2$:

$$\min_{\Theta} L(\Theta, D) = \frac{1}{N} \sum_n l(\theta_1, \ldots, \theta_K; x_n) + \lambda_1 \sum_k \alpha(\theta_k) + \lambda_2 h(B), \tag{1}$$

# 5 METHOD

In the Method section, we first introduce the framework of VFedCD in 5.1, including how to approximate the causal mechanism through encoder-decoder. Then we introduces SDP in 5.2, which is presented prior to the CTV because the redesigning of feature encoding and aggregation in SDP fundamentally influences the graph extraction process in CTV. In 5.3, we explain how CTV aggregates causal graphs and enforces structural constraints. The overall approach is shown in Fig. 2.

The pseudo-code can be found in the Appendix A. Complete proofs for the computational complexity and communication overhead are provided in Appendices B and C, including detailed assumptions and derivations. For identifiable details, please refer to Appendix E.

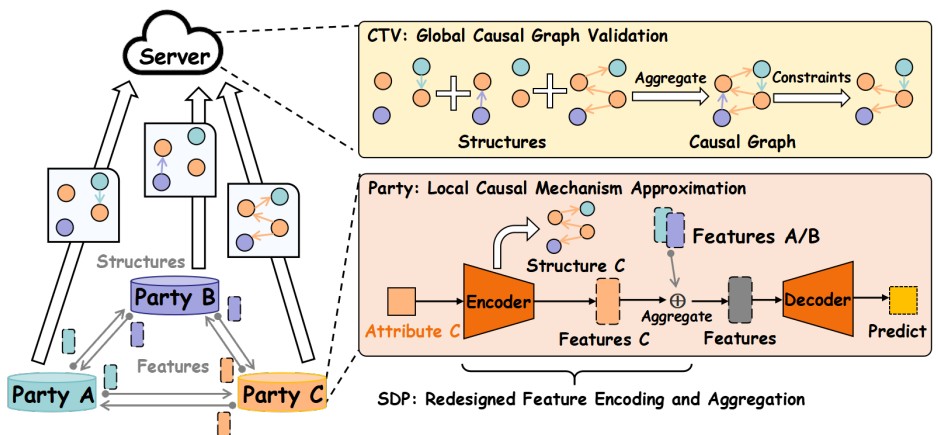

Figure 2: VFedCD approach. The left side shows the communication between parties and CTV. The upper right is CTV. The lower right is the expansion of party C, and SDP redesigning feature encoding and aggregation.

## 5.1 FRAMEWORK

VFedCD enables each party to collaboratively approximate causal mechanisms and learn causal graph structures through encoder–decoder networks. For clarity, we decompose the method into two stages: forward propagation and backward propagation, as outlined below.

**Forward.** The forward propagation process, mainly visualized in Party in Fig. 2, unfolds in three steps:

- **Step1: Encoding.** Each party $k$ encodes its local attributes $x_n^k$ into $K$ feature vectors $\{H_{kt}\}_{t=1}^K$ using $K$ sub-encoders $\{F_{kt}\}_{t=1}^K$. The sub-encoder $F_{kt}$, parameterized by $\phi_{kt}$, generates $H_{kt} = F_{kt}(x_n^k; \phi_{kt})$, representing the causal contribution of party $k$'s data to the prediction process of target party $t$. This decomposes the encoder into specialized components for cross-party feature generation.

- **Step 2: Cross-Party Feature Transmission.** For a target party $t$ predicting its attributes, all parties $k = 1, \ldots, K$ transmit the feature $H_{kt}$ (specifically designed for target $t$) to party $t$. This step enables each party to collect the features it needs from all other parties, forming a collaborative feature pool for subsequent decoding.

- **Step 3: Decoding.** Target party $t$ aggregates the received features $\{H_{kt}\}_{k=1}^K$ and uses its decoder $G_t$ (parameterized by $\omega_t$) to reconstruct its local attributes:

$$\hat{x}_n^t = G_t(H_{1t}, H_{2t}, \ldots, H_{Kt}; \omega_t), \tag{2}$$

The decoder maps the aggregated features to the attribute space, approximating the causal mechanisms that generate the target party's data from its causal parents across all parties.

**Backward.** During backward propagation, target party $t$ first computes the prediction loss $l_{con}(\hat{x}_n^t, x_n^t)$ between the reconstructed attributes $\hat{x}_n^t$ and the true data $x_n^t$. The gradient $\nabla_{l_{con}} \hat{x}_n^t$ of reconstruction loss $l_{con}$ is then backpropagated through the decoder $G_t$ to compute gradients with respect to the intermediate features $\{H_{kt}\}_{k=1}^K$:

$$\nabla_{l_{con}} H_{kt} = \frac{\partial l_{con}}{\partial H_{kt}} \quad k = 1, \ldots, K, \tag{3}$$

These gradients are transmitted back to the respective source parties $k$, which update their sub-encoder parameters $\phi_{kt}$ using stochastic gradient descent (SGD) in a VFL-compatible manner (Yang et al., 2019). The decoder parameters $\omega_t$ are updated locally by party $t$ to minimize the reconstruction error.

## 5.2 Secure Dispatch Protocol (SDP)

In traditional VFL with deep-encoder shallow-decoder (dEsD) architectures, feature encryption is often applied during decoding, where lightweight shallow decoders can handle the computational overhead of homomorphic operations. However, transitioning to a shallow-encoder deep-decoder (sEdD) architecture for improved inter-party causal mechanism modeling introduces a critical challenge: deep decoders' complex nonlinear operations make traditional full homomorphic encryption (FHE) computationally infeasible. Additionally, using raw data as "labels" for loss computation creates dual leakage paths. Based on the features and gradients received, party can respectively inference other parties' data and "labels". Since raw data plays the role of "label", adversaries can cross - validate its inference of other parties' sensitive data from features and gradients.

Inspired by (Fu et al., 2022), we design a encryption strategy to align with the sEdD architecture. As shown in lower right of Fig. 2, SDP focuses on securing the encoding and feature aggregation processes rather than the decoding stage. By decomposing feature computations into encrypted fragments via semi-homomorphic encryption and secret sharing, SDP ensures that no party can reconstruct standalone features or gradients during training, while avoiding the computational burden of encrypting deep decoder operations. The detailed threat model can be found in Appendix H.

SDP consists of three components: initialization for key and model setup, forward propagation for secure feature aggregation, and backward propagation for privacy-preserving gradient updates. Without loss of generality, the following uses the prediction of attributes in target party $p_t$ as an example to introduce the protocol. The prediction of attributes in other parties follows the same process.

**Initialization.** As shown in the Algorithm 2, each party $p_k$ generates a key pair $\langle p_k.pub, p_k.pri \rangle$ and initializes $K$ plaintext encoders $\{p_k.f_{it}\}$, parameterized by $\phi_{kit}$. Here, $p_i.f_{kt}$ represents a fragment of the complete encoder $F_{kt}$ (which maps $x_k$ to features for target party $t$) that is distributed to party $i$. Specifically, the complete encoder $F_{kt}$ is decomposed into $K$ fragments across all parties, i.e., $F_{kt}(x_k) = \sum_{i=1}^K p_i.f_{kt}(x_k)$. Thus, the features $Z_t$ required for predicting $x_t$ can be formulated as:

$$Z_t = \sum_{k=1}^K F_{kt}(x_k) = \sum_{k=1}^K \sum_{i=1}^K p_i.f_{kt}(x_k), \tag{4}$$

Notably, direct computation of $Z_t$ via this formula is infeasible because $x_k$ resides exclusively on party $k$, while its corresponding encoder fragments $p_i.f_{kt}$ are distributed across all parties $i$. The subsequent steps of SDP (forward and backward propagation) are designed to indirectly achieve the computation of $Z_t$ on target party $t$ without exposing raw data or complete encoder parameters.

Then, for $j \neq k$, party $p_k$ encrypts $p_k.f_{jt}$ with its public key $p_k.pub$ as $\|p_k.f_{jt}\|_k$ and sends it to party $j$, which stores it as a encrypted $\|p_j.ef_{kt}\|_k$.

**Forward.** As shown in the Algorithm 3, for target party $p_t$, each party $p_k$ computes: 1. **Plaintext feature**: $p_k.f_{kt}(x_k)$ (direct contribution from party $k$ to itself). 2. **Ciphertext features**: $\|p_k.ef_{it}(x_k)\|_i = x_k^\top \|p_k.ef_{it}\|_i$ (for $i \neq k$), transformed via HE2SS into secret shares $[\epsilon_{kit}, remaining_{kit}]_{k,i}$. Party $p_k$ aggregates these as $z_{kt} = p_k.f_{kt}(x_k) + \sum_{i \neq k}(\epsilon_{kit} + remaining_{ikt})$ and sends $z_{kt}$ to party $p_t$, which sums all $z_{kt}$ to get $Z_t = \sum_k z_{kt}$ for decoding.

**Backward.** As shown in the Algorithm 4, Target party $p_t$ computes gradients $\nabla Z_t$ from the decoder, encrypts them as $\|\nabla Z_t\|_t = E(\nabla Z_t, p_t.pub)$, and broadcasts to other parties. Each party $k \neq t$ first computes $\|x_k^\top \nabla Z_t\|_t = x_k^\top \|\nabla Z_t\|_t$ and applies HE2SS to divide into $[\varphi_{kt}, grad\_remaining_{kt}]_{k,t}$. And then, it updates its encoder $p_k.f_{kt}$ using $\varphi_{k,t}$, while party $p_t$ updates $p_t.f_{kt}$ using $grad\_remaining_{kt}$. Finally, the updated encoders are re-encrypted and exchanged to maintain model consistency across parties.

## 5.3 Centralized Topology Validator (CTV)

In vertically partitioned settings, each party's local subgraph structures is embedded within encoder parameters, thus no single party can verify the global graph's topological validity. Without centralized validation, the inferred graph may contain cycles or overly dense connections, violating causal discovery's fundamental requirements. While transmitting both features and local subgraphs between parties to enforce constraints raises significant privacy risks, thus not contributing a solution. To address this, we introduce the CTV, a server-based component that aggregates encoder parameters to validate the global graph's acyclicity using structural constraints, as shown in CTV in Fig. 2. Verifying global topological validity without exposing raw graph structure data, CTV balances constraint enforcement and privacy preservation. We detail the forward process to obtain a subgraph with the attributes of party $p_t$ as the effect, and the backward process to update the corresponding models.

**Forward.** As shown in Algorithm 5, each party $p_k$ firstly extracts a causal graph structure from its encoders $B_t^k = graph(\{p_k.f_{it}\}_{i=1}^K)$ and secondly sends to the CTV. Thirdly, the CTV constructs the subgraph $B_t = \sum_{k=1}^K B_t^k$. Note that the structural constraints is enforced on full graph $B = concat(\{B_t\}_{t=1}^K)$, which is gained by repeating the process for all $K$ target parties. To ensure the validity of the complete causal graph, these constraints typically include two types: acyclicity and sparsity.

To enforce acyclicity, the CTV leverages the spectral radius constraint $h(B)$ (Nazaret et al., 2023), defined as:

$$h(B) = \rho(B) = \max_{1 \leq i \leq d} |\lambda_i(B)|, \tag{5}$$

where $\lambda_i(B)$ denotes the $i$-th eigenvalue of $B$. This constraint ensures the graph is a directed acyclic graph (DAG). Beyond acyclicity, structural constraints may also incorporate sparsity-promoting terms (e.g., the $\ell_1$-regularization $\|B\|_1$) to penalize overly dense connections, though their implementation details are omitted here for focus. The combined structural objective (e.g., $\mathcal{L}_{\text{struct}} = \lambda_2 \|B\|_1 + \lambda_3 h(B)$) guides the learning process toward both valid and interpretable causal graph structures. We provide further details on acyclicity constraints in the appendix D.

**Backward.** As shown in the Algorithm 6, to update models for predicting attributes on target party $p_t$, the CTV computes and randomly splits the gradient $\nabla_h B_t = \sum_{k=1}^K grad\_structure_{kt}$, where $grad\_structure_{kt}$ are gradients sent to party $p_k$. Parties further decompose received gradients into encoder-specific updates

$$grad\_structure_{kit} = chunk(grad\_structure_{kt}), \tag{6}$$

for updating $p_k.f_{it}$, aligning local parameter adjustments with the global acyclicity constraint without exposing full graph gradients.

## 6 Experiments

### 6.1 Experimental Setup

**Datasets.** We evaluate VFedCD on both synthetic and real-world datasets. Synthetic datasets are generated with varying numbers of attributes (10, 15, 25) and edge densities (30 to 75 edges) to test scalability and robustness. For real-world validation, we use the Sachs (Sachs et al., 2005), SynTReN (Van den Bulcke et al., 2006), and a diabetes (Kahn) dataset. All centralized datasets are artificially partitioned to simulate the VFL setting. We use an 80/20 train/test split for all datasets.

**Implementation Details.** We use Stochastic Gradient Descent (SGD) with a learning rate of 0.01, a batch size of 16 and 500 training epochs. By default, we set the regularization hyperparameter $\lambda_1$ to $5 \times 10^{-3}$, and communication cycle $Q$ to 1. The acyclicity constraints weight $\lambda_2$ starts at 0 and increases by $gamma$ (0.006) per epoch, with its growth halted once the current graph becomes acyclic. All experiments are conducted on a server equipped with 8×V100 GPUs.

**Evaluation Metrics.** For datasets with known ground-truth causal graphs, we adopt two standard evaluation metrics:

- **Structural Hamming Distance (SHD)** (de Jongh & Druzdzel, 2009), which measures the number of edge additions, deletions, or reversals needed to convert the inferred graph into the true directed acyclic graph (DAG).

- **F1 Score**, which evaluates the accuracy of edge detection by harmonizing precision and recall.

To evaluate privacy protection, we implement *Unsplit attacks* (Erdoğan et al., 2022), where a semi-honest party attempts to infer other participants' raw data. We report the **Absolute Correlation** between the inferred features and the true data, with lower values indicating stronger privacy preservation. For the diabetes dataset, which does not have a ground-truth causal graph, we conduct a qualitative evaluation by examining whether the inferred causal relationships align with established medical knowledge.

**SOTA methods.** Some SOTA methods addressing causal discovery for learning causal graph with centralized data are included: NO-TEARS (Zheng et al., 2018), NO-BEARS (Lee et al., 2019), DAGMA (Bello et al., 2022) , Sortnregress (Reisach et al., 2021), DCDI (Brouillard et al., 2020), DCD-FG (Lopez et al., 2022). Detailed introduction to these methods is included in Appendix F.

## 6.2 Performance Comparison on Synthetic Data

This section demonstrates VFedCD's competitive performance against centralized causal discovery methods under vertical partitioning. Table 1 compares SHD across synthetic datasets with different number of attributes. Despite not requiring raw data sharing, VFedCD achieves comparable results to centralized approaches. We repeat Each experiment five times and take the mean value. For seed-sensitive methods, the standard deviation of F1 is marked.

Table 1: Causal discovery accuracy on synthetic datasets. The red asterisk (*) indicates methods with complexity higher than $O(D^2)$. Bolded values represent the best two performances.

| Attributes | **10 Attributes** | | **15 Attributes** | | **25 Attributes** | |
|---|---|---|---|---|---|---|
| Metric | SHD ↓ | F1 ↑ | SHD ↓ | F1 ↑ | SHD ↓ | F1 ↑ |
| NO-TEARS* | 29 | 0.433 | 62 | 0.110 | 84 | 0.272 |
| NO-BEARS | 36.4 | $0.205_{\pm 0.052}$ | 57.2 | $0.265_{\pm 0.023}$ | 96.8 | $0.075_{\pm 0.026}$ |
| DAGMA* | 29.4 | $0.362_{\pm 0.175}$ | **42.2** | $\mathbf{0.539_{\pm 0.023}}$ | **77** | $0.393_{\pm 0.075}$ |
| Sortnregress* | 31 | 0.355 | 54 | 0.267 | 102 | 0.300 |
| DCDI* | 30 | $0.405_{\pm 0.071}$ | 52.4 | $0.306_{\pm 0.041}$ | **67.2** | $\mathbf{0.607_{\pm 0.037}}$ |
| DCD-FG | **19.2** | $\mathbf{0.632_{\pm 0.082}}$ | 61.8 | $0.446_{\pm 0.060}$ | 192.5 | $0.244_{\pm 0.037}$ |
| VFedCD(Ours) | **18.8** | $\mathbf{0.698_{\pm 0.016}}$ | **33** | $\mathbf{0.711_{\pm 0.013}}$ | 81.6 | $\mathbf{0.649_{\pm 0.008}}$ |

## 6.3 Generalization

To evaluate model robustness, we test VFedCD on datasets with varying characteristics: Synthetic Dataset (15 attributes, edge counts: 30, 45, 60, 75), Real-World Dataset (Sachs protein network), Highly Realistic Synthetic Dataset (SynTReN dataset). Table 2 presents results for seven methods across these scenarios. VFedCD demonstrates consistent performance across different graph structures. Unbalanced data partitioning will not have a significant impact on the results, but it will affect the training time. For details about unbalanced data partitioning, please refer to the appendix G.

Furthermore, to demonstrate the universality of VFedCD, we conduct an experiment on a practical diabetes dataset without real causal graph and presented the consistency between the predicuted causal graph and clinical knowledge. Details are provided in Appendix I.

Table 2: Generalization performance across diverse datasets (SHD ↓ F1 ↑).

| | Synthetic Datasets(SHD) | | | | Sachs | SynTReN |
|---|---|---|---|---|---|---|
| | 30 Edges | 45 Edges | 60 Edges | 75 Edges | SHD | F1 |
| NO-TEARS | **21** | 38 | 62 | 68 | **17** | 0.256 |
| NO-BEARS | 25 | 40 | 59 | 69 | **17** | 0.201 |
| DAGMA | 26 | **21** | **45** | 63 | 18 | **0.277** |
| Sortnregress | 39 | 40 | 54 | 67 | **16** | 0.257 |
| DCDI | 29 | 38 | 50 | 60 | 38 | 0.136 |
| DCD-FG | 73 | 86 | 64 | **57** | 40 | 0.168 |
| VFedCD(Ours) | **13** | **27** | **32** | **31** | 19 | **0.407** |

## 6.4 ABLATION STUDY

We evaluate VFedCD components on a synthetic dataset with 15 attributes. A semi-honest party conducts Unsplit attacks (Erdoğan et al., 2022) to infer others' raw data. We compare three variants: (1) the framework, (2) the framework + CTV, and (3) the full VFedCD. Metrics include SHD, F1 - score, and the Absolute Correlation between inferred and raw features.

As Table 3 shows, CTV reduces SHD by 35.7% through global topology validation. With SDP, the correlation drops 62%, indicating strong privacy protection. Attackers obtain only weakly correlated features (0.152), proving the protocol's effectiveness against semi-honest adversaries. The analysis of collusion attacks when the semi-honest assumption is broken and the further improvement of security policies leveraging differential privacy are presented in the appendix H.

Table 3: Ablation study (15-attribute dataset). Corr.: Absolute Correlation of Unsplit attacks.

| Method | SHD ↓ | F1 ↑ | Corr. ↓ |
|---|---|---|---|
| Framework | 56 | 0.610 | 0.443 |
| Framework+CTV | 36 | 0.678 | 0.400 |
| VFedCD | **35** | **0.700** | **0.152** |

## 6.5 ARCHITECTURE ANALYSIS

We explore how the vertical model architecture affects edge recovery using a synthetic dataset with 15 attributes. In this context, we define two types of edges: inter-party edges and intra-party edges, classified based on whether the causes of edge's target attribute are within a party or distributed across different parties. Fig. 3 visualizes the impact of different encoder - decoder configurations on these two types of edges. The left subfigure shows various architecture setups with different encoder/decoder depths. The right compares their F1 - scores for two edge types.

Results show that the dEsD architecture amplifies the F1 gap between intra and inter edges. This indicates that the dEsD architecture is poor at modeling inter-party dependencies, as it focuses more on intra-party edges. In contrast, VFedCD's sEdD architecture (Split Position A) minimizes this disparity. This demonstrates that the sEdD architecture is necessary in achieving a balanced discovery of both intra-party and inter-party causal mechanisms. We provide an intuitive example in the appendix E.5 to understand this difference.

## 6.6 ROBUSTNESS STUDY

This section evaluates VFedCD's stability against hyperparameters and communication cycle. We analyze the parameter $\gamma$ (acyclicity constraint coefficient $\lambda_2$, linearly increasing $\gamma$ per epoch when

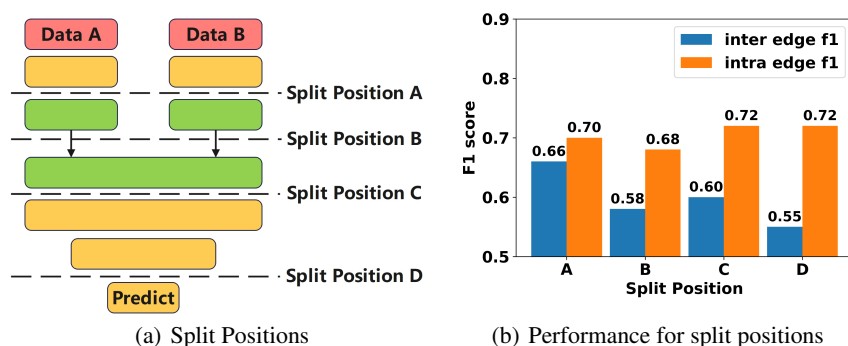

(a) Split Positions        (b) Performance for split positions

Figure 3: Splitting Strategy Analysis

current graph is not acyclic) and $Q$ (communication cycle, following FedBCD's design (Liu et al., 2022), where parties communicate every $Q$ updates).

As Fig. 4 shows, varying $\gamma$ between 0.004 and 0.007 causes minimal SHD/F1 fluctuations, indicating insensitivity to this constraint strength. Increasing $q$ from 1 to 5 reduces the final F1 score from 0.72 to 0.53, yet the model remains functional, demonstrating robustness to reduced communication efficiency. These results confirm VFedCD's practical viability in diverse settings.

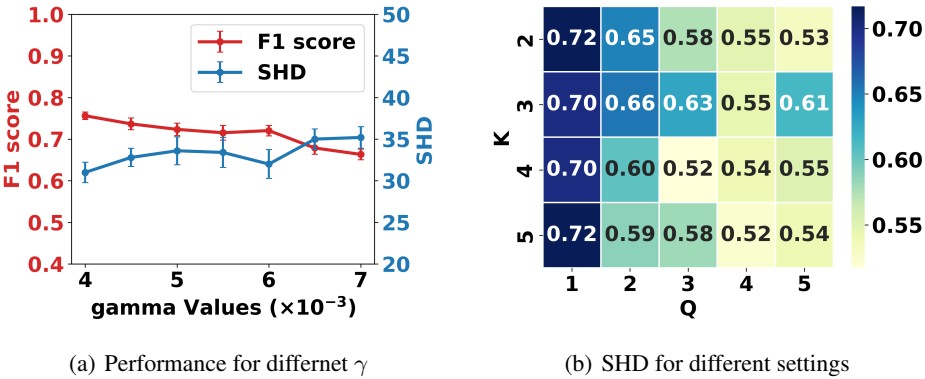

(a) Performance for differnet $\gamma$        (b) SHD for different settings

Figure 4: Robustness Analysis to Hyper Parameter

## 7   CONCLUSION AND LIMITATIONS

**Conclusion.** In this work, we establish a theoretical foundation for causal discovery in vertically federated learning. We propose VFedCD, the first framework tailored for vertical federated causal discovery, incorporating two key components: CTV, which enforces global acyclicity constraints, and SDP, which enables privacy-preserving feature interactions. Both experimental results and theoretical analyses validate the effectiveness of VFedCD in terms of causal discovery validity, privacy protection, and computational efficiency. Specifically, VFedCD ensures causal discovery validity through global acyclicity enforcement and balanced edge modeling, safeguards privacy against both data and label inference, and achieves practical computational efficiency via lightweight encryption.

**Limitations.** However, several limitations need further discussion. First, scalability to extremely high-dimensional data remains a challenge, requiring more efficient cryptographic optimization. Second, strategies to reduce communication overhead have not been extensively explored. Third, real federated deployments require mechanisms for fair contribution assessment and benefit distribution, ensuring parties providing valuable data receive proportional rewards. Future work may address these limitations while preserving the core strengths of the proposed framework.

## REPRODUCIBILITY STATEMENT

We take reproducibility seriously and have provided the necessary details to facilitate replication of our results. Appendix A contains detailed pseudo-code for both forward and backward propagation, outlining the key steps of our proposed framework. In the supplementary materials, we include the complete pipeline for implementing VFedCD, including pre-processing, model training, and evaluation. For the synthetic datasets used in this study, we also provide the data within the supplementary materials. Additionally, all hyperparameter settings and experimental configurations are explicitly described in Section 6.

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

## A  PSEUDO CODE

**Algorithm 1** The algorithm to transform an HE variable $\|data\|_A$ into an SS variable $[\epsilon, data - \epsilon]_{A,B}$

---

1: **procedure** HE2SS($p_B$, $p_A.pub$, $\|data\|_A$)
2:     $\epsilon = random\_noise()$
3:     $\|data + \epsilon\|_A = \|data\|_A + E(-\epsilon, p_A.pub)$
4:     $send\_to(p_B, \|data - \epsilon\|_A)$
5:     $return\ \epsilon$
6: **end procedure**
7: **procedure** HE2SS($p_A$, $p_B.pri$)
8:     $\|data - \epsilon\|_B = receive\_from(p\_A)$
9:     $(data - \epsilon) = decrypt(\|data - \epsilon\|_B, p_B.pri)$
10:    $remaining = (data - \epsilon)$
11:    $return\ remaining$
12: **end procedure**

---

**Algorithm 2** Initialization algorithm

---

1: **procedure** INITIALIZATION OF PARTY K
2:     $(p_k.pub, p_k.pri) = generate\_key\_pair()$
3:     **for** $i = 1$ to $K$ **do**
4:         $p_k.f_{it} = initialize\_plaintext\_model()$
5:     **end for**
6:     **for** $j = 1$ to $K$ **do**
7:         **if** $j \neq k$ **then**
8:             $\|p_k.f_{jt}\|_k = E(p_k.f_{jt}, p_k.pub)$
9:             $send\_to(p_j, \|p_k.f_{jt}\|_k)$
10:            $\|p_k.ef_{jt}\|_j = receive\_from(p_j)$
11:         **end if**
12:     **end for**
13: **end procedure**

---

**Algorithm 3** Forward algorithm for $Z_t$ in SDP

---

1: **procedure** FORWARD OF PARTY K($x_k$)
2:     $component_{kt} = x_k^T(p_k.f_{kt})$
3:     **for** $i = 1$ to $K$ **do**
4:         **if** $i \neq k$ **then**
5:             $\|p_k.ef_{it}(x_k)\|_i = x_k^T\|p_k.ef_{it}\|_i$
6:         **end if**
7:     **end for**
8:     **for** $i = 1$ to $K$ **do**
9:         **if** $i \neq k$ **then**
10:            $\varepsilon_{kit} = HE2SS(p_i, p_k.pub, \|p_k.ef_{it}(x_k)\|_i)$
11:            $remaining_{ikt} = HE2SS(p_i, p_k.pri)$
12:         **end if**
13:     **end for**
14:    $z_{kt} = component_{kt} + \sum_{i=1,i\neq k}^{K} \varepsilon_{kit} + \sum_{i=1,i\neq k}^{K} remaining_{ikt}$
15:    $send\_to(p_t, z_{kt})$
16: **end procedure**
17: **procedure** FORWARD OF TARGET PARTY T
18:    $Z_t = 0$
19:    **for** $k = 1$ to $K$ **do**
20:        **if** $k \neq t$ **then**
21:            $z_{kt} = receive\_from(p_k)$
22:            $Z_t = Z_t + z_{kt}$
23:        **end if**
24:    **end for**
25: **end procedure**

**Algorithm 4** Backward algorithm from $x^t$ in SDP

1: **procedure** BACKWARD OF PARTY K
2:     $\|\nabla Z_t\|_t = receive\_from(p_t)$
3:     $\|x_k^T \nabla Z_t\|_t = x_k^T \|\nabla Z_t\|_t$
4:     $\varphi_{kt} = HE2SS(p_t, p_k.pub, \|x_k^T \nabla Z\|_t)$
5:     $p_k.f_{kt} = update\_model(p_k.f_{kt}, \varphi_{kt})$
6:     $\|p_k.ef_{tt}\|_t = receive\_from(p_t)$
7: **end procedure**
8: **procedure** BACKWARD OF TARGET PARTY T($\nabla Z_t$)
9:     $\nabla F_{tt} = X_t^T \nabla Z_t$
10:     $p_t.f_{tt} = update\_model(p_t.f_{tt}, \nabla F_t)$
11:     $\|\nabla Z_t\|_t = E(\nabla Z_t, p_t.pub)$
12:     **for** $k = 1$ to $K$ **do**
13:         **if** $k \neq t$ **then**
14:             $send\_to(p_k, \|\nabla Z_t\|_t)$
15:         **end if**
16:     **end for**
17:     **for** $k = 1$ to $K$ **do**
18:         **if** $k \neq t$ **then**
19:             $remaining\_grad_{kt} = HE2SS(p_k, p_t.pri)$
20:             $p_t.f_{kt} = update\_model(p_t.f_k, remaining\_grad_{kt})$
21:             $\|p_t.f_{kt}\|_t = E(p_t.f_{kt}, p_t.pub)$
22:             $send\_to(p_k, \|p_t.f_{kt}\|_t)$
23:         **end if**
24:     **end for**
25: **end procedure**

**Algorithm 5** Forward algorithm for $B_t$ in CTV

1: **procedure** FORWARD OF PARTY K
2:     $B_{kt} = graph(\{p_k.f_{it}\})$
3:     $send\_to(CTV, B_{kt})$
4: **end procedure**
5: **procedure** FORWARD OF CTV
6:     **for** $k = 1$ to $K$ **do**
7:         $B_{kt} = receive\_from(p_k)$
8:     **end for**
9:     $B_t = aggregate(\{B_{kt}\}_{k=1}^K)$
10: **end procedure**

**Algorithm 6** Backward algorithm from $B_t$ in CTV

1: **procedure** BACKWARD FOR PARTY K
2:     $grad\_structure_{kt} = receive\_from(CTV)$
3:     $\{grad\_structure_{kit}\} = chunk(grad\_structure_{kt})$
4:     $p_k.f_{it} = update\_model(p_k.f_{it}, grad\_structure_{kit})$
5: **end procedure**
6: **procedure** BACKWARD FOR CTV($\nabla_h B^t$)
7:     $\{grad\_structure_{kt}\}_{k=1}^K = random\_split(\nabla_h B_t)$
8:     **for** $k = 1$ to $K$ **do**
9:         $send\_to(p_k, grad\_structure_{kt})$
10:     **end for**
11: **end procedure**

# B   COMPUTATIONAL COMPLEXITY ANALYSIS OF VFEDCD

The computational complexity of the VFedCD is dominated by homomorphically encrypted (HE) matrix multiplications. We analyze the complexity by mapping operations to their corresponding pseudocode steps.

## B.1   FORWARD PROPAGATION COMPLEXITY

For a target party $t$, each party $k$ computes $K - 1$ HE matrix multiplications $\{\|p_k.ef_{jt}(x_k)\|\}_{j\neq k}$ (Algorithm 3, line 5). Each multiplication has complexity $O(d_k d_t)$, where $d_k$ and $d_t$ are the attribute dimensions of party $k$ and $t$, respectively.

The total complexity for target party $t$ is:

$$\sum_{k=1}^{K}(K-1)O(d_k d_t)$$

Extending this to all $K$ target parties, the overall forward-pass complexity becomes:

$$\sum_{t=1}^{K}\sum_{k=1}^{K}(K-1)O(d_k d_t)$$

## B.2   BACKWARD PROPAGATION COMPLEXITY

For backward propagation from target party $t$, each party $k \neq t$ performs the HE matrix multiplication $x_k^T\|\nabla Z_t\|_t$ (Algorithm 4, line 3) with complexity $O(d_k d_t)$. The total complexity for target party $t$ is:

$$\sum_{k=1,k\neq t}^{K} O(d_k d_t)$$

Summing over all target parties, the total backward complexity is:

$$\sum_{t=1}^{K}\sum_{k=1,k\neq t}^{K} O(d_k d_t)$$

## B.3   OVERALL COMPLEXITY

The dominant term combines forward and backward complexities:

$$\sum_{t=1}^{K}\sum_{k=1}^{K}(K-1)O(d_k d_t) + \sum_{t=1}^{K}\sum_{k=1,k\neq t}^{K} O(d_k d_t)$$

Assuming uniform attribute distribution ($d_k = D/K$), this simplifies to $O(KD^2)$, where $D$ is the total attribute dimension.

## B.4   TRAINING TIME ANALYSIS

We first characterize the theoretical complexity: with synchronization constraints, the training time is bounded by the party with the maximum number of attributes $d_{\max} = \max_k d_k$, with a worst-case per-party complexity of $O(d_{\max}D)$. Beyond theoretical analysis, we aim to provide a *generalizable framework* for practical scalability assessment, using a baseline hardware configuration to quantify feasible boundaries—this framework can be easily adapted to different hardware setups (e.g., multi-core CPUs, GPUs) by adjusting the benchmarked parameters.

**Empirical Benchmark as a Reference Baseline**   To establish a foundational reference, we measured HE multiplication performance under a simple, non-parallelized setup: a single-core AMD R9-7945HX CPU using Python's Paillier homomorphic encryption library. This setup yields 12,500 HE multiplications per second. For practicality, we define a feasible iteration as one completing within 30 minutes, with HE operations (the dominant cost) allocated the full 30-minute budget. This gives a maximum of $30 \times 60 \times 12,500 = 22,500,000$ HE multiplications per iteration as our baseline constraint.

**Scalability Calculation** For uniform attribute distribution ($d_k = d/K$) and typical hyperparameters (hidden_dim=10), we derive the total HE multiplications per iteration by combining forward and backward propagation costs (detailed in Algorithms 3 and 4): - Forward propagation: Each party performs $(K-1) \cdot n \cdot (d/K) \cdot d \cdot 10$ HE multiplications. - Backward propagation: Each party performs $(d/K) \cdot n \cdot d \cdot 10$ HE multiplications.

Simplifying these, the total HE multiplications per iteration scale as $10 \cdot n \cdot d^2$, where $n$ is the sample count and $d$ is the total number of attributes.

**Feasibility Boundaries Under Baseline Setup** Using the single-core benchmark, we quantify feasible scales within the 30-minute iteration budget: - *Typical scenario*: For $n = 1000$ samples and $d = 15$ attributes, 2.25 million HE operations are required, completing in 180 seconds (well within budget). - *Moderate scale*: For $n = 800$ samples and $d = 50$ attributes, 20 million HE operations are needed, taking 1600 seconds (27 minutes, near the limit). - *Beyond feasibility*: For $n = 1000$ samples and $d = 50$ attributes, 25 million HE operations exceed the budget, requiring 2000 seconds (33 minutes).

These boundaries ($d < 50$, $n < 1000$ under the single-core setup) cover most causal discovery datasets, but importantly, they reflect hardware constraints rather than algorithmic limitations.

**Extending to Other Hardware Configurations** The core advantage of our framework is its adaptability to diverse hardware. Since our matrix multiplication-based encrypted operations are inherently parallelizable, the runtime scales with the number of parallel processing units. For $M$ parallel CPUs/GPUs, the effective runtime reduces to $O(nd^2/M)$. Readers can thus: 1. Benchmark their own hardware to determine HE multiplications per second. 2. Apply the same $10 \cdot n \cdot d^2$ formula to calculate feasible $n$ and $d$ for their specific 30-minute (or other) budget.

This design ensures our scalability analysis remains generalizable, providing a clear path for adapting VFedCD to various computational environments.

# C COMMUNICATION OVERHEAD ANALYSIS

## C.1 FORWARD PROPAGATION OVERHEAD

For a single target party $t$, the communication overhead is as follows: 1. **Share Collection**: Each party $k$ collects $K - 1$ shares $remaining_{ikt}$ from other parties (Algorithm 3, lines 10–11). The per-party overhead is $(K-1)O(d_t)$, leading to a total of $K(K-1)O(d_t)$ across all $K$ parties. 2. **Result Transmission**: All parties $k \neq t$ send $z_{kt}$ to $t$ (Algorithm 3, lines 15, 21), contributing $(K-1)O(d_t)$.

Extending to all $K$ target parties, the total party-to-party overhead becomes:

$$\sum_{t=1}^{K} [K(K-1)O(d_t) + (K-1)O(d_t)] = O(K^2D).$$

In addition, each party needs to send the corresponding parameters to the server. Considering the model for computing the intermediate results required by party $t$, the data volume of the parameters sent by party $k$ is $O(Dd_t)$. The total communication complexity for all parties to send parameters for computing the intermediate results required by party $t$ is $KO(Dd_t)$. When extended to all parties $t$ (i.e., all models), the data volume of the transmitted parameters is $KO(D^2)$, which is $O(KD^2)$.

## C.2 BACKWARD PROPAGATION OVERHEAD

For a single target party $t$: 1. **Gradient Broadcast**: Party $t$ encrypts and broadcasts $\|\nabla Z_t\|_t$ to $K - 1$ parties (Algorithm 4, lines 2, 14), incurring $(K-1)O(d_t)$. 2. **Gradient Share Transmission**: Each party $k \neq t$ sends $remaining\_grad_{kt}$ to $t$ (Algorithm 4, lines 4, 19). The total overhead across all $k \neq t$ is $\sum_{k \neq t} O(d_k d_t) = O(Dd_t)$. 3. **Model Update**: Party $t$ sends updated encrypted models $\|p_t.f_{kt}\|_t$ to all $k \neq t$ (Algorithm 4, line 22), with total overhead $O(Dd_t)$.

Summing over all $K$ target parties, the party-to-party overhead is:

$$\sum_{t=1}^{K} [(K-1)O(d_t) + O(Dd_t) + O(Dd_t)] = O(KD + D^2).$$

The server backpropagates each parameter gradient sent during the forward propagation. So, this part of the communication overhead is the same as the communication overhead of parties sending parameters to the server during the forward propagation, which is $O(KD^2)$.

### C.3 OVERALL COMMUNICATION OVERHEAD

Combining all components:

Forward: $O(K^2D)$ + Backward: $O(K^2D + D^2)$ + Server: $O(KD^2) = O(K^2D + KD^2)$.

## D GRAPH FUNCTION AND ACYCLICITY CONSTRAINT CALCULATION

### D.1 FROM MODEL PARAMETERS TO CONTINUOUS ADJACENCY MATRIX

The function graph$(\Theta)$ transforms model parameters into a continuous adjacency matrix, quantifying the strength of potential causal relationships. It operates in two steps: 1. **Extracting Linear Layer Weights**: For each encoder-decoder pair targeting party $t$, the first linear layer parameters (denoted $\theta_{kt} \in \mathbb{R}^{d_k \times (d_t \cdot h)}$, where $h$ is the hidden dimension) are extracted. These parameters directly encode the influence of local attributes (from party $k$) on features for the target party $t$. 2. **Norm Aggregation**: The hidden dimension $h$ is aggregated using the $L_2$ norm, resulting in a matrix $B \in \mathbb{R}_{\geq 0}^{d_k \times d_t}$. Each entry $B_{ij}$ represents the continuous strength of the causal edge from attribute $i$ (party $k$) to attribute $j$ (party $t$), with larger values indicating stronger inferred causality.

### D.2 ACYCLICITY CONSTRAINT CALCULATION

To enforce the acyclic property of the global causal graph, we adopt the spectral acyclicity constraint from Nazaret et al. (2023), which leverages the spectral radius of the adjacency matrix. The spectral radius is defined as:

$$h_\rho(B) = |\lambda_{\max}(B)|$$

where $\lambda_{\max}(B)$ is the largest eigenvalue (in magnitude) of the adjacency matrix $B$.

This constraint exhibits key properties that make it suitable for measuring cyclicity: - For a directed acyclic graph (DAG), its adjacency matrix $B$ is acyclic, meaning no cycles exist in the graph. By graph theory, acyclic matrices are nilpotent (i.e., there exists a positive integer $k$ such that $B^k = 0$) . A fundamental property of nilpotent matrices is that all their eigenvalues are zero . Thus, for a DAG, $\lambda_{\max}(B) = 0$, so $h_\rho(B) = 0$. - For cyclic graphs, the adjacency matrix $B$ contains at least one cycle. Such matrices are not nilpotent and must have at least one non-zero eigenvalue . Consequently, $h_\rho(B) > 0$, where larger values indicate stronger cyclicity. This is because cycles with heavier edge weights or more nodes contribute to larger eigenvalue magnitudes .

During training, the acyclicity loss is incorporated as $\mathcal{L}_{\text{acyclic}} = \gamma \cdot h_\rho(B)$, where $\gamma$ is a scaling factor. Minimizing this loss encourages the model to learn adjacency matrices with $h_\rho(B) \approx 0$, thus promoting acyclic structures.

## E IDENTIFIABILITY ANALYSIS

Identifiability of causal graphs in vertical federated learning requires the framework to uniquely recover the true causal structure from observed data. This section provides a rigorous theoretical analysis, establishing how our framework achieves identifiability through the Additive Noise Model (ANM) and faithfulness assumptions, complemented by architectural design.

### E.1 FUNDAMENTAL DEFINITIONS AND IDENTIFIABILITY CRITERION

We begin by formalizing key concepts and the core requirement for identifiability.

### E.1.1 KEY SETS AND NOTATIONS

For each variable $X_i \in \mathcal{D}$ (where $\mathcal{D} = \{X_1, X_2, \ldots, X_d\}$ is the set of all observed variables), define:
- $C_i^{\text{true}} \subseteq \mathcal{D} \setminus \{X_i\}$: The set of *true direct causes* of $X_i$ (i.e., $X_j \in C_i^{\text{true}} \iff X_j \to X_i$ in the true causal graph). - $E_i^{\text{true}} \subseteq \mathcal{D} \setminus \{X_i\}$: The set of *true direct effects* of $X_i$ (i.e., $X_k \in E_i^{\text{true}} \iff X_i \to X_k$ in the true causal graph). - $I_i^{\text{true}} \subseteq \mathcal{D} \setminus \{X_i\}$: The set of variables with *no direct causal link* to $X_i$ (neither cause nor effect).

By definition, these sets are mutually exclusive and exhaustive:

$$\mathcal{D} \setminus \{X_i\} = C_i^{\text{true}} \cup E_i^{\text{true}} \cup I_i^{\text{true}}, \quad \text{and } C_i^{\text{true}} \cap E_i^{\text{true}} = C_i^{\text{true}} \cap I_i^{\text{true}} = E_i^{\text{true}} \cap I_i^{\text{true}} = \emptyset.$$

Let $C_i^{\text{pred}} \subseteq \mathcal{D} \setminus \{X_i\}$ denote the set of variables *predicted as direct causes* of $X_i$ by VFedCD.

### E.1.2 IDENTIFIABILITY CRITERION

A causal discovery framework is identifiable if and only if, for *all* $X_i \in \mathcal{D}$:

$$C_i^{\text{pred}} = C_i^{\text{true}}.$$

This equality requires three conditions to hold simultaneously: 1. *Inclusion of true causes*: $C_i^{\text{true}} \subseteq C_i^{\text{pred}}$ (no true cause is omitted). 2. *Exclusion of true effects*: $E_i^{\text{true}} \cap C_i^{\text{pred}} = \emptyset$ (no effect is mistaken for a cause). 3. *Exclusion of irrelevant variables*: $I_i^{\text{true}} \cap C_i^{\text{pred}} = \emptyset$ (no causally unrelated variable is included).

The remainder of this section demonstrates how ANM and faithfulness assumptions ensure these three conditions in VFedCD.

## E.2 ROLE OF THE ADDITIVE NOISE MODEL (ANM) ASSUMPTION

The ANM assumption states that for each $X_i \in \mathcal{D}$, the true causal mechanism follows:

$$X_i = f_i\left(C_i^{\text{true}}\right) + \epsilon_i, \quad \text{where } \epsilon_i \perp\!\!\!\perp C_i^{\text{true}},$$

where $f_i$ is a measurable function (capturing the causal mechanism), and $\epsilon_i$ (noise) is statistically independent of $C_i^{\text{true}}$. This assumption enables VFedCD to distinguish $C_i^{\text{true}}$, $E_i^{\text{true}}$, and $I_i^{\text{true}}$ through residual analysis and loss minimization.

### E.2.1 ENSURING INCLUSION OF TRUE CAUSES ($C_i^{\text{TRUE}} \subseteq C_i^{\text{PRED}}$)

Suppose a true cause $X_j \in C_i^{\text{true}}$ is mistakenly excluded from $C_i^{\text{pred}}$. By ANM, $X_i$ depends on $X_j$ through $f_i$, so the residual of $X_i$ predicted using $C_i^{\text{pred}}$ (denoted $\epsilon_i(C_i^{\text{pred}}) = X_i - \hat{f}_i(C_i^{\text{pred}})$) will correlate with $X_j$:

$$\epsilon_i(C_i^{\text{pred}}) \not\perp\!\!\!\perp X_j.$$

This correlation implies the reconstruction loss will be larger than if $X_j$ were included:

$$\left\| \epsilon_i(C_i^{\text{pred}}) \right\|^2 > \left\| \epsilon_i(C_i^{\text{pred}} \cup \{X_j\}) \right\|^2.$$

Since VFedCD minimizes this loss, the model is incentivized to include $X_j$ in $C_i^{\text{pred}}$. For all $X_j \in C_i^{\text{true}}$, this ensures $C_i^{\text{true}} \subseteq C_i^{\text{pred}}$.

### E.2.2 ENSURING EXCLUSION OF TRUE EFFECTS ($E_i^{\text{TRUE}} \cap C_i^{\text{PRED}} = \emptyset$)

Variables $X_k \in E_i^{\text{true}}$ are correlated with $X_i$ (due to $X_i \to X_k$) but are not causes of $X_i$. To avoid mistaking $X_k$ for a cause: - VFedCD enforces acyclicity via constraints that prohibit both $X_i \to X_k$ and $X_k \to X_i$ (i.e., $B_{ik} > 0 \implies B_{ki} = 0$ for adjacency matrix $B$). - When $C_i^{\text{true}} \subseteq C_i^{\text{pred}}$, $X_k \in E_i^{\text{true}}$ is independent of $\epsilon_i(C_i^{\text{pred}})$ (by ANM). Including $X_k$ in $C_i^{\text{pred}}$ reduces loss only weakly (via overfitting), while correctly modeling $X_i \to X_k$ yields a significant loss reduction (since $\epsilon_k \perp\!\!\!\perp C_k^{\text{true}}$ by ANM for $X_k$).

This dynamic pushes $B_{ki} \to 0$, ensuring $E_i^{\text{true}} \cap C_i^{\text{pred}} = \emptyset$.

### E.2.3 Ensuring Exclusion of Irrelevant Variables ($I_i^{\text{TRUE}} \cap C_i^{\text{PRED}} = \emptyset$)

Variables $X_l \in I_i^{\text{true}}$ have no causal link to $X_i$ but may coincidentally reduce reconstruction loss. However: - By ANM, when $C_i^{\text{true}} \subseteq C_i^{\text{pred}}$, $\epsilon_i(C_i^{\text{pred}}) \perp\!\!\!\perp \mathcal{D} \setminus \{X_i\}$, so including $X_l$ reduces loss only through overfitting (magnitude far smaller than including a true cause). - VFedCD's $L_1$ sparsity regularization ($\lambda_2 \|B\|_1$) penalizes non-zero entries for irrelevant variables, pushing $B_{li} \to 0$.

Together, these ensure $I_i^{\text{true}} \cap C_i^{\text{pred}} = \emptyset$.

### E.3 Role of the Faithfulness Assumption

The faithfulness assumption ensures that *all conditional independencies in the data are exactly those implied by d-separation in the true causal graph*. Formally:

$$X_i \perp\!\!\!\perp X_j \mid S \iff X_i \text{ and } X_j \text{ are d-separated by } S \text{ in the true graph}$$

for any $X_i, X_j \in \mathcal{D}$ and $S \subseteq \mathcal{D} \setminus \{X_i, X_j\}$.

This assumption is critical because it guarantees that dependencies captured by VFedCD's loss function and graph constraints *genuinely reflect causal relationships*, not spurious correlations from hidden confounders. Without faithfulness: - Latent variables $L$ (where $L \to X_i$ and $L \to X_j$) create $X_i \not\!\perp\!\!\!\perp X_j \mid \emptyset$ even if $X_i$ and $X_j$ are d-separated, leading to false edges $X_i \to X_j$ or $X_j \to X_i$. - Such spurious correlations corrupt the residual independence properties relied on by ANM, weakening the model's ability to distinguish $C_i^{\text{true}}$, $E_i^{\text{true}}$, and $I_i^{\text{true}}$.

In VFedCD, faithfulness ensures that the ANM-based mechanisms (Section 3.2) operate on "clean" dependencies, preserving the three conditions for identifiability.

### E.4 Empirical Validation of Assumptions

#### E.4.1 ANM Violation Experiments

We test performance under noise models that violate ANM properties (Table 4):

| Noise Type | Violated Property | SHD ↓ | F1 ↑ |
|---|---|---|---|
| Additive (ANM) | None | 108 | 0.638 |
| Multiplicative | Additivity | 109 | 0.628 |
| Poisson | Additivity | 64 | 0.683 |
| Non-iid Additive | Independence | 123 | 0.612 |

Table 4: Performance Under ANM Violations (30-attribute dataset)

Non-iid additive noise (violating independence) degrades performance most, confirming independence is critical. Multiplicative noise (violating additivity) performs comparably, aligning with our theoretical focus on independence over additivity.

#### E.4.2 Faithfulness Violation Experiments

Testing with latent confounders (Table 5):

| Scenario | SHD ↓ | F1 ↑ |
|---|---|---|
| 25 attributes (no latents) | 53 | 0.654 |
| 20 attributes (5 latents) | 70 | 0.537 |

Table 5: Performance Under Faithfulness Violations

Latent confounders increase SHD by 32%, but partial recovery persists, demonstrating robustness to real-world violations.

### E.5 Edge Type Identifiability

In vertical federated settings, intra-party edges (causes and effects within one party) and inter-party edges (causes across multiple parties) exhibit different identifiability properties, largely dependent on the encoder-decoder architecture.

#### E.5.1 Example Setup

Consider a causal relationship $x^C = (x^A - 0.5) \cdot (x^B)^2$, where $x^A \in$ Party A, $x^B \in$ Party B, and $x^C \in$ Party C. We generate training samples:

$$(x^A, x^B) = (0.0, 1.0) \implies x^C = -0.5; \quad (x^A, x^B) = (1.0, 1.0) \implies x^C = 0.5$$

#### E.5.2 dEsD Architecture Limitation

Deep encoders: $z^A = F_A(x^A) = \sigma(W_A \sigma(\phi_A x^A))$, $z^B = F_B(x^B) = \sigma(W_B \sigma(\phi_B x^B))$.

Shallow Decoder: $\hat{x}^C = G(z_A, z_B) = (z_A + z_B)/2$.

Given MSE loss $L = (\hat{x}^C - x^C)^2$, the gradient with respect to $z^B$ is:

$$\nabla_{z^B} L = (\hat{x}^C - x^C)$$

Considering network initialization with near-zero parameters and activation functions centered at zero (e.g., $\tanh$), we analyze SGD updates:

1. **Sample 1**: $(x^A, x^B) = (0.0, 1.0)$
   - Initial prediction $\hat{x}^C \approx 0$ (from near-zero $z^A, z^B$)
   - Residual error: $0 - (-0.5) = +0.5 \implies \nabla_{z^B} L = +0.5$
   - Update direction: $\phi_B \leftarrow \phi_B - \eta \cdot (+0.5)$
2. **Sample 2**: $(x^A, x^B) = (1.0, 1.0)$
   - Initial prediction $\hat{x}^C \approx 0$
   - Residual error: $0 - 0.5 = -0.5 \implies \nabla_{z^B} L = -0.5$
   - Update direction: $\phi_B \leftarrow \phi_B - \eta \cdot (-0.5)$

This creates **conflicting gradients** for identical input $x^B = 1.0$:

- Positive gradient (+0.5) from Sample 1 pushes $\phi_B$ to decrease
- Negative gradient (-0.5) from Sample 2 pulls $\phi_B$ to increase

The zero-mean activation function converts this conflict into gradient cancellation:

$$\mathbb{E}[\nabla_{\phi_B} L] = \frac{1}{2}(+0.5) + \frac{1}{2}(-0.5) = 0$$

causing $\phi_B$ to converge near zero, thereby suppressing the $B \to C$ causal edge identification.

#### E.5.3 sEdD Architecture Correctness

Encoders: $z^A = F_A(x^A) = \phi_A x^A$, $z^B = F_B(x^B) = \phi_B x^B$.

Decoder: $G(z_A, z_B) = \sigma(W\sigma(z^A + z^B))$.

The Universal Approximation Theorem (Hornik et al., 1990) guarantees existence of $G$ such that:

$$G(w_A x^A + w_B x^B) \approx (x^A - 0.5) \cdot (x^B)^2 \tag{7}$$

Gradients align coherently as $\frac{\partial G}{\partial z_B}$, captures the dependency on $x^B$, allowing $\phi_B$ to remain non-zero and recover the $B \to C$ edge.

#### E.5.4 Empirical Validation

sEdD achieves balanced F1 scores (0.70 for intra-party vs. 0.66 for inter-party edges), while dEsD shows a large gap (0.72 vs. 0.55), confirming sEdD's superiority for inter-party edge identifiability.

### E.6 Conclusion

Identifiability in VFedCD is theoretically guaranteed by: 1. **ANM**: Enabling distinction between $C_i^{\text{true}}$ (via residual independence), $E_i^{\text{true}}$ (via acyclicity), and $I_i^{\text{true}}$ (via sparsity). 2. **Faithfulness**: Ensuring data dependencies reflect true causality, so ANM-based distinctions remain valid. 3. **sEdD Architecture**: Preserving inter-party edge identifiability in federated settings.

Together, these components ensure $C_i^{\text{pred}} = C_i^{\text{true}}$ for all $X_i$, achieving causal graph identifiability.

## F SOTA METHODS

### F.1 INTRODUCTION TO SOTA METHODS

All SOTA methods address causal discovery for learning causal graph with centralized data. NOTEARS (Zheng et al., 2018) converts DAG learning into a continuous optimization problem using matrix exponential trace constraints and augmented Lagrangian methods with L1 regularization. DAGMA (Bello et al., 2022) uses a log-determinant acyclicity constraint with M-matrices and a central path optimization to detect large cycles efficiently and improve computational speed. DCD-FG (Lopez et al., 2022) introduces factor DAGs (f-DAGs) with low-rank structures and Gaussian nonlinear models, scaling to high-dimensional interventional data via GPU acceleration. DCDI (Brouillard et al., 2020) employs differentiable frameworks with normalizing flows to handle perfect/imperfect interventions, maximizing log-likelihood to identify Markov equivalence classes. NO-BEARS (Lee et al., 2019) enhances NOTEARS by replacing matrix exponentials with spectral radius approximations and adding polynomial regression for nonlinear data, leveraging GPU speedups. Sortnregress (Reisach et al., 2021) acts as a baseline by sorting variables by marginal variance and using sparse regression, exposing how data scaling and varsortability influence benchmark performance.

### F.2 COMPARISON WITH CENTRALIZED SOTA METHODS

Our experimental results show that VFedCD, a distributed method, achieves performance comparable or even superior to some centralized SOTA methods on certain datasets. This outcome stems not from an inherent superiority in all conditions, but from the alignment of its modeling assumptions with the characteristics of the data, particularly our nonlinear synthetic datasets.

VFedCD is built upon general assumptions, namely the Additive Noise Model (ANM) and Faithfulness, which allow it to flexibly model a wide range of nonlinear causal mechanisms. In contrast, many centralized methods are designed with more specific assumptions or for particular scenarios:

Methods with Linearity Assumptions: The original formulations of NOTEARS (Zheng et al., 2018) and its scalable successor NO-BEARS (Lee et al., 2019), as well as DAGMA (Bello et al., 2022), are primarily based on linear structural equation models. Consequently, their performance is naturally limited on datasets with complex nonlinear relationships, where VFedCD's nonlinear modeling capacity provides a distinct advantage.

Methods for Specific Data Types or Structures: DCDI (Brouillard et al., 2020) excels in its sophisticated handling of various interventional data types through normalizing flows. DCD-FG (Lopez et al., 2022) is specifically designed to scale to very high-dimensional settings by assuming a low-rank factor graph structure. While powerful in their respective niches, their designs are not universally optimal for general observational data.

Baselines Revealing Methodological Artifacts: Sortnregress (Reisach et al., 2021) serves as a crucial baseline, demonstrating that many methods may inadvertently perform well by exploiting the "varsortability" of data, rather than by capturing the true causal structure.

Therefore, VFedCD's strong performance on our synthetic data is a direct consequence of its robust, general-purpose nonlinear modeling, which is well-suited for the data generation process. It highlights that in the diverse landscape of causal discovery, there is no single best method, but rather a trade-off between generality and specialization. VFedCD's contribution lies in successfully bringing a general and powerful modeling paradigm into the challenging, privacy-constrained vertical federated setting.

## G   IMPACT OF UNBALANCED DATA PARTITION

To address concerns about heterogeneous vertical partitioning where parties hold imbalanced numbers of features, we conduct supplementary experiments on a 15-attribute synthetic dataset with 3 parties, using three specific attribute distribution configurations to systematically evaluate the impact: [1, 1, 13], [3, 3, 9], and [5, 5, 5]. These configurations exhibit different levels of imbalance, quantified by attribute variance.

For each configuration, we measure two key metrics: the Structural Hamming Distance (SHD, lower is better) and F1 score (higher is better) of VFedCD, which reflect the accuracy of causal discovery. The results are summarized in Table 6.

Table 6: Performance of VFedCD under Different Imbalanced Attribute Distributions

| Data Partition | Attribute Variance | SHD $\downarrow$ | F1 $\uparrow$ |
|---|---|---|---|
| $[1, 1, 13]$ | 32 | 36 | 0.68 |
| $[3, 3, 9]$ | 8 | 35 | 0.69 |
| $[5, 5, 5]$ | 0 | 35 | 0.70 |

As shown in Table 6, the F1 score remains stable (ranging from 0.68 to 0.70) and SHD shows minimal fluctuation across all imbalance levels, indicating that **imbalanced attribute distribution does not introduce significant bias** in VFedCD. This robustness stems from our shallow-encoder deep-decoder (sEdD) architecture, which is specifically designed to handle cross-party causal mechanisms. Even when attributes are unevenly distributed, the deep decoder effectively aggregates features from all parties, ensuring balanced modeling of both intra-party and inter-party causal relationships.

We also observe that training time increases with imbalance severity: when the maximum number of attributes held by a single party reaches 13 (in the [1, 1, 13] configuration), training time increases by 82% compared to the balanced [5, 5, 5] partitioning. This aligns with our computational complexity analysis in Appendix B, where larger $d_{max}$ (maximum number of attributes per party) incur higher costs.

These results suggest that while VFedCD maintains causal discovery performance under imbalanced partitioning, practical deployment should consider resource allocation strategies for parties with heavy computational loads, especially in synchronous training settings.

## H   PRIVACY SECURITY ANALYSIS IN SDP

This section formalizes the threat model for VFedCD, analyzes potential privacy vulnerabilities, and evaluates mitigation strategies against collusion attacks.

### H.1   THREAT MODEL AND VULNERABILITIES

We define the adversary model and identify key privacy risks, focusing on scenarios where semi-honest assumptions may be violated.

#### H.1.1   ADVERSARY MODEL

We consider two primary types of adversaries: 1. **Semi-honest parties**: Parties strictly follow the protocol but attempt to infer other parties' raw data using observed information (e.g., aggregated features, encrypted parameters). 2. **Colluding among parties**: A subset of parties colluded, the situation of privacy leakage has hardly intensified. 3. **Colluding with CTV**: A subset of parties colluding with the CTV server, leveraging plaintext causal graph structures to enhance inference capabilities.

#### H.1.2   VULNERABILITIES

1. **Feature leakage**: Shallow encoders generate linear features that retain statistical correlations with raw data. Unlike deep encoders, which abstract data into high-level representations, shallow

encoders' outputs may reveal patterns exploitable via inference attacks. 2. **Violation of semi-honest assumptions**: While single parties or small groups of colluding parties have limited inference power (as shown in Section 3, with correlation $\leq 0.152$), collusion between a party and the CTV poses a significant risk. The CTV receives plaintext graph fragments $B_t^k$, which are strongly correlated with encoder parameters, providing structural insights that complement the party's local information (e.g., aggregated features, decoder parameters). This combination enables more accurate data reconstruction than single-party attacks. Gradient leakage is not a concern here, as gradients are protected via secret sharing (Section 5.2), preventing direct access to sensitive information.

## H.2 SINGLE-PARTY INFERENCE ATTACKS

We analyze the feasibility of Unsplit attacks (Erdoğan et al., 2022) by a single semi-honest party, detailing inference bounds and practical limitations.

### H.2.1 INFERENCE BOUND

A single party $p_t$ attempting to infer $p_k$'s data ($k \neq t$) employs the Unsplit attack, which formulates data reconstruction as an optimization problem:

$$\min_{\{\bar{F}_{kt}, \bar{x}_n^k\}} \left\| \sum_{k=1}^{K} \bar{F}_{kt}(\bar{x}_n^k) - Z \right\|_2^2$$

where $\bar{F}_{kt}$ is an auxiliary encoder with the same architecture as $F_{kt}$, and $\bar{x}_n^k$ are the adversary's guesses for $p_k$'s raw data.

The optimal solution to this problem exhibits two critical properties: 1. **Proportional scaling**: For any non-zero $\sigma$, solutions satisfying $\bar{\psi}_{kt} = \sigma\psi_{kt}$, $\bar{\xi}_{kt} = \xi_{kt}$, and $\bar{x}_n^k = \sigma^{-1}x_n^k$ minimize the loss. This implies the adversary can only reconstruct data up to a scalar multiple of the true values. 2. **Non-permutability**: Permuting features (e.g., swapping elements of $\bar{x}_n^k$ and corresponding rows of $\bar{\psi}_{kt}$) does not minimize the loss due to masked self-loop constraints in causal discovery (enforcing $\psi[i,i] = 0$).

These properties bound the maximum inferable precision, with the absolute correlation coefficient $|\text{Corr}(\bar{x}, x)|$ serving as a valid metric (upper bound = 1.0).

### H.2.2 PRACTICAL VERIFICATION

The above analysis shows the maximum inferable precision is a proportional scaling of the true data, without permutations. This makes the absolute correlation coefficient $|\text{Corr.}|$ (Ali Abd Al-Hameed, 2022) a valid metric to quantify inference risk, as it reaches its maximum at inference upper bound.

**Complexity Barrier** Let $d_t$ be the target attribute dimension and $m$ the decoder depth. Each training epoch provides $d_t m$ equations but requires solving for:

$$\underbrace{K(d_k m_1 d_t)}_{\text{encoder weights}} + \underbrace{K d_k}_{\text{auxiliary data}} = O(Dm_1 d_t) \text{ unknowns} \tag{8}$$

where $D = \sum d_k$. The underdetermined system grows exponentially with $K$, rendering exact inference computationally difficult.

**Empirical Validation** The SDP reduces feature correlation to 0.152, far below the theoretical upper bound (1.0). This gap confirms the combinatorial explosion prevents adversaries from approaching proportional scaling solutions.

## H.3 COLLUSION ATTACK: INTER-PARTY COLLUSION

### H.3.1 THREAT ANALYSIS

Collusion among multiple parties poses limited risk due to two fundamental barriers: 1. **Encoder parameter opacity**: Parties lack knowledge of encoder weights $\{\psi_{kt}\}$, which are critical for reverse-engineering raw data. 2. **Encrypted feature fragmentation**: Data is distributed and encrypted,

requiring colluders to solve an underdetermined system with more unknowns (encoder params + target data) than equations. The more parties involved collude, the more unknowns there will be.

### H.3.2 EMPIRICAL VALIDATION

We evaluate collusion between 3 parties (out of 4 total) on a 30-attribute dataset with partition [8,8,7,7]:

| Scenario | Inference Correlation |
|---|---|
| Single-party inference | 0.163 |
| 3-party collusion | 0.179 |

Table 7: Inference Performance Under Inter-Party Collusion

The marginal increase (0.179 vs. 0.163) confirms minimal benefits from collusion, as the combinatorial complexity of solving for encoder parameters dominates over additional data fragments.

### H.4 COLLUSION ATTACK: PARTY + CTV

Collusion between a party and the CTV is the most severe threat, as the CTV's graph structures complement the party's local information. We detail this attack and its mitigation.

### H.4.1 THREAT OF COLLUSION

The CTV aggregates local graph fragments $B_t^k$ to enforce acyclicity, providing colluding parties with explicit structural information about causal relationships. This allows adversaries to: 1. Fix encoder architectures to match $B_t$, reducing the number of unknowns in the Unsplit optimization. 2. Cross-validate inferred data with graph structure (e.g., ensuring edges in $B_t$ align with correlations in reconstructed data).

Experimental validation on a $K = 2$, $d = 20$ dataset shows collusion enables $|\text{Corr}| = 0.804$ (far exceeding single-party performance).

### H.4.2 MITIGATION WITH DIFFERENTIAL PRIVACY

To counter collusion, we apply differential privacy to graph fragments $B_t^k$ before transmission to the CTV, using a Laplace mechanism:

$$\hat{B}_t^k = B_t^k + Lap(b), \quad b = \frac{\Delta}{\epsilon}$$

where $\Delta = 1$ (sensitivity) and $\epsilon$ controls the privacy-utility tradeoff.

As shown in Figure 5: 1. Without noise: $|\text{Corr}| = 0.804$, SHD=42 (high leakage, optimal utility). 2. $\epsilon = 256$: $|\text{Corr}| = 0.660$, SHD=44 (moderate leakage, minimal utility loss). 3. $\epsilon = 128$: $|\text{Corr}| = 0.458$, SHD=46 (low leakage, acceptable utility). 4. $\epsilon = 64$: $|\text{Corr}| = 0.119$, SHD=47 (negligible leakage, moderate utility loss).

These results indicate $\epsilon = 64$ or $128$ balances privacy and utility effectively.

### H.4.3 SCALING NOISE FOR MULTIPLE PARTIES

VFedCD avoids the inherent limitation of local differential privacy (LDP), where noise accumulates with the number of parties $K$, degrading utility. In our framework: - Graph fragments $B_t^k$ are meaningless in isolation; only their aggregation $B_t = \sum B_t^k$ represents the true causal structure. - Thus, we only need to protect the aggregated $B_t$, not individual $B_t^k$.

To maintain stable noise impact on $B_t$ as $K$ increases, we scale the Laplace parameter $b$ proportional to $1/\sqrt{K}$ (proof in H.6). For $K = 2$ with $b = 0.0078125$ (base case, $\epsilon = 128$), $K = 4$ uses $b = 0.0078125/\sqrt{2} \approx 0.0055$, ensuring consistent privacy guarantees.

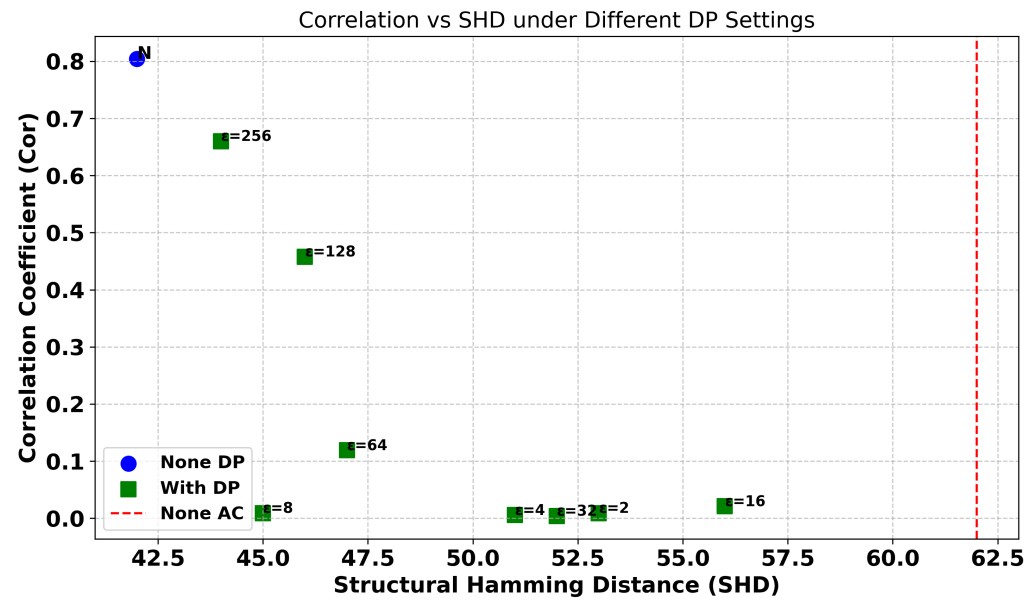

Figure 5: Privacy-utility tradeoff under party-CTV collusion with Laplace noise. Baselines: SHD=35 with no noise (None DP), SHD=62 without acyclicity constraint (None AC).

## H.5 CONCLUSION

SDP thwarts single-party attacks via complexity barriers. For collusion, Laplace noise with $\epsilon = 64$ or 128 for $K = 2$, scaled by $1/\sqrt{K}$, ensures robust privacy while preserving causal discovery utility.

## H.6 LAPLACE NOISE SCALING PROOF FOR MULTIPLE PARTIES

### H.6.1 CONCLUSION

For $k$ independent and identically distributed (IID) Laplace random variables with mean 0 and scale parameter $b$, let $S$ denote their sum, and let $m = \mathbb{E}[|S|]$ be the expectation of the absolute value of $S$. To maintain the stability of $m$ (i.e., keep $m$ relatively unchanged) when the number of variables $k$ changes, the scale parameter $b$ should be adjusted proportionally to $1/\sqrt{k}$. This means $b \propto 1/\sqrt{k}$.

### H.6.2 PROOF

**Properties of a Single Laplace Variable** A Laplace random variable $X$ with mean 0 and scale parameter $b$ has the probability density function (PDF):

$$f_X(x) = \frac{1}{2b} \exp\left(-\frac{|x|}{b}\right), \quad x \in \mathbb{R}$$

It can be decomposed into the difference of two independent exponential variables: $X = Y - Z$, where $Y, Z \sim \text{Exp}(1/b)$ (exponential distribution with rate $1/b$) and $Y \perp\!\!\!\perp Z$.

**Sum of $k$ IID Laplace Variables** Let $X_1, X_2, \ldots, X_k$ be IID Laplace variables with $X_i \sim \text{Laplace}(0, b)$. Their sum is:

$$S = X_1 + X_2 + \cdots + X_k$$

Using the decomposition $X_i = Y_i - Z_i$ for each $i$, we rewrite $S$ as:

$$S = \left(\sum_{i=1}^{k} Y_i\right) - \left(\sum_{i=1}^{k} Z_i\right) = U - V$$

where: 1. $U = \sum_{i=1}^{k} Y_i \sim \text{Gamma}(k, b)$ (sum of $k$ exponential variables), 2. $V = \sum_{i=1}^{k} Z_i \sim \text{Gamma}(k, b)$ (sum of $k$ exponential variables), 3. $U$ and $V$ are independent.

**Expectation of $|S|$**   The expectation $m = \mathbb{E}[|S|] = \mathbb{E}[|U - V|]$ is derived using properties of the Gamma distribution. For independent $U, V \sim \text{Gamma}(k, b)$, the expectation simplifies to:

$$m = \frac{2b}{\sqrt{\pi}} \cdot \frac{\Gamma\left(k + \frac{1}{2}\right)}{\Gamma(k)}$$

where $\Gamma(\cdot)$ is the Gamma function.

**Scaling Strategy for $b$**   To keep $m$ stable when $k$ changes, we analyze the behavior of $m$ with respect to $k$. For large $k$, the Gamma function satisfies the Stirling approximation:

$$\Gamma(z) \approx \sqrt{2\pi} z^{z - \frac{1}{2}} e^{-z} \quad \text{for large } z.$$

Applying this to $\Gamma\left(k + \frac{1}{2}\right) / \Gamma(k)$ gives:

$$\frac{\Gamma\left(k + \frac{1}{2}\right)}{\Gamma(k)} \approx \sqrt{k} \quad \text{for large } k.$$

Substituting into $m$, we get:

$$m \approx \frac{2b}{\sqrt{\pi}} \cdot \sqrt{k}.$$

To keep $m$ unchanged when $k$ increases by a factor $n$ (i.e., $k \to nk$), $b$ must scale as:

$$b \propto \frac{1}{\sqrt{k}}.$$

Specifically, if $k$ is multiplied by $n$, $b$ should be divided by $\sqrt{n}$ (i.e., $b \to b/\sqrt{n}$) to maintain $m \approx$ constant.

**Error of the Approximation**   The approximation $b \propto 1/\sqrt{k}$ improves as $k$ increases, because the Stirling formula becomes more accurate for large $k$. For small $k$, the error is slightly larger but remains bounded (as shown in prior analysis, typically ¡ 15% even for $k = 1$).

Thus, scaling $b$ with $1/\sqrt{k}$ ensures $m$ remains relatively stable for multiple parties (i.e., increasing $k$).

## I   PRACTICAL IMPLEMENTATION VERIFICATION

We demonstrate VFedCD's capability on a practical diabetes dataset with vertical partitioning. Party A, representing a public healthcare system, has demographic and basic health metrics such as Age, Glucose Levels, and Diabetes Diagnosis. Party B, a specialized clinic, contributes advanced diagnostic features like Pregnancies, Insulin levels, and Diabetes Pedigree Function.

As shown in Fig. 6, a local causal graph inferred by VFedCD. The model accurately identifies Pregnancies and Diabetes Pedigree Function from Party B as causes of diabetes, in line with medical knowledge on genetic and reproductive risk factors. It also correctly determines that blood glucose from Party A and insulin levels from Party B are downstream effects of diabetes, consistent with islet cell dysfunction pathophysiology. This verifies VFedCD's practicability in data-siloed scenarios.

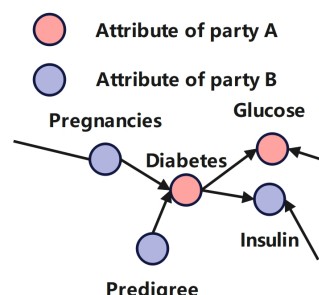

Figure 6: Local causal graph with some key attributes related to diabetes.

## J  THE USE OF LARGE LANGUAGE MODELS (LLMs)

The contributions of Large Language Models (LLMs) to this paper were limited to non-critical tasks such as language refinement, debugging portions of the code, and generating simple scripts (e.g., plotting utilities). The research ideas, theoretical development, draft preparation, and primary pipeline implementation were carried out entirely by the authors without LLM involvement. The core methodology, experimental design, and all key contributions are the sole intellectual work of the authors.

