# OpenReview forum: "VFedCD: Causal Discovery under Vertical Federated Scenario"
_ICLR.cc/2026/Conference — ICLR 2026 Conference Withdrawn Submission_

### Official Review · Reviewer_8Ljo · 2025-10-18

**Soundness:** 2
**Presentation:** 2
**Contribution:** 2
**Rating:** 2
**Confidence:** 5

**Summary:**

This paper studies causal discovery in the VFL setting, where different parties hold disjoint feature sets for the same individuals. It proposes **VFedCD**, a split-model framework with a coordinating component that enforces a continuous acyclicity constraint and a cross-party feature/representation transmission mechanism to enable joint DAG learning without raw-data pooling. The manuscript claims three contributions: (1) a system-level VFL pipeline for causal structure learning, (2) an algorithm that optimizes a continuous DAG objective in a distributed fashion, and (3) theoretical and empirical validation of the approach on synthetic and benchmark datasets.

**Strengths:**

- **Timely topic.**
  The paper studies causal discovery in VFL, which is indeed a relevant direction given increasing data privacy concerns across domains such as healthcare and finance.

- **Conceptual motivation.**
  The authors make an effort to bridge the areas of causal discovery and federated learning, highlighting an underexplored intersection that could be of potential interest to both communities.

- **Interesting idea.**
  The overall workflow, dividing models across parties with a coordination component, is conceptually understandable, and the manuscript provides a basic sketch of how such a system might be implemented.

**Weaknesses:**

- **Poor organization and unclear notations.**
  The paper is written in a confusing manner, with inconsistent and frequently redefined notations throughout. Key symbols (e.g., **B**, **Θ**, **D**) change their meanings across sections, making it very difficult to follow the methodology or reproduce the setup. This lack of clarity in mathematical definitions and notation usage seriously undermines readability and technical credibility.

- **Unclear problem definition and data formulation.**
  The problem statement of VFedCD is not well specified. It remains ambiguous what exact task the model is solving, how the global causal graph is defined, and how the data are partitioned among clients. The paper does not clearly describe whether nodes can overlap between parties or how information flow is constrained under the vertical FL setting. As a result, it is unclear what precise learning objective the proposed method is supposed to optimize.

- **Questionable federated learning design.**
  The proposed *Cross-Party Feature Transmission* mechanism seems to directly share intermediate representations or feature information across clients. This violates the core privacy assumption of federated learning, where no raw or derived data should be transmitted between parties. Without a proper explanation or privacy-preserving mechanism, the proposed framework cannot be considered a valid federated solution.

- **Lack of theoretical justification and identifiability discussion.**
  The paper claims to “establish a theoretical foundation” but provides no formal analysis or guarantees. There is no discussion of under what assumptions the proposed approach can recover the true DAG, nor what identifiability conditions are required. Without any theoretical results or guarantees, the method remains purely heuristic and its validity is questionable.

**Questions:**

- The authors claim that this is the first *vertical federated causal discovery* framework. However, to my knowledge, the work *“Horizontal and Vertical Federated Causal Structure Learning via Higher-order Cumulants”* already addresses similar settings and was proposed several months ago. In addition, the authors state that they “establish a theoretical foundation” for **VFedCD**, but it is unclear what specific theory is provided — where is it presented or formally proven?

- In the *Related Works* section, many citations and categorizations appear to be incorrect or inconsistent.
  - The **PC** algorithm was originally proposed by *Peter and Clark*, not by *Kalisch and Bühlmann*.
  - **GES** is cited under both score-based and mechanism-fitting methods, which is confusing.
  - **NOTEARS** is incorrectly excluded from score-based methods — it searches over DAGs via continuous optimization and thus should belong to that category.
  - The term *“mechanism-fitting branches”* is unclear. From my understanding, the community typically classifies *LiNGAM*, *ANM*, and *PNL* as **constrained functional causal models**.
  - There seems to be a typo in “NOTEARS-ADMMTh method.”
  - The notations are inconsistent: for example, **D** and **B** are first defined as functions for different parties, but later redefined as datasets and causal graphs.

- The *Problem Setup* section is rather confusing and lacks clarity on the overall formulation.
  - What exactly is the causal graph — is it assumed to be a DAG?
  - What is the underlying data generation process, and how is it partitioned among clients?
  - Can nodes overlap between clients?
  - The “learning objective” claims that the ground-truth causal graph can be obtained via a loss function, which sounds more like a *solution* rather than a *problem definition*.
  - The relationship between **B** and **Θ** is unclear.
  - What is the formulation of the function **h**, and how does it enforce the continuous acyclicity constraint?
  - How is the loss **l** defined, and how does it aggregate information from multiple parameters and the global observation **x**?
  - The term **φ** is also unclear — why does *t* range from 1 to *K*? Is **φ** an encoder specific to client *k*?

- In the *Method* section:
  - The proposed **Cross-Party Feature Transmission** mechanism appears to transmit feature information across clients, which contradicts the privacy-preserving assumption in federated learning. Please clarify how this complies with the federated setup.
  - In the backward process, how is the prediction loss defined — mean squared error or something else?
  - In Eq. (5), what is the role of the additional notation **ρ**? Why is it needed?

- The authors should carefully proofread the manuscript before submission. There are many inconsistent abbreviations and typographical issues.
  - For example, the full name of **SGD** is introduced both at line 224 and again at line 324.
  - The acronym **DAG** is not defined until line 335.
  - It is confusing to refer to all baseline methods as *SOTA*; most of them are not designed for **VFedCD**. How are these baselines adapted to the proposed setting?
  - How is the synthetic dataset constructed?
  - Finally, why is subsection 6.3 titled “Generalization”? The term seems misused in this context.

---

> ### Author Response · Authors · 2025-11-15
>
> We thank Reviewer for the constructive feedback. We address the major concerns below:
>
> ## Regarding Weakness 1 (Notation Clarity), Weakness 2 (Problem Definition), Q2-Q5 (Specific Questions):
>
> We will thoroughly revise notation and problem formulation: (1) **Add symbol table** defining all notations (B, Θ, D, K, etc.) and subscripts ($B^k$, $B_{kt}$). (2) **Rewrite Problem Setup** clarifying: task (learn global DAG $B \in \mathbb{R}^{D \times D}$ from vertically partitioned data), data partition (party k holds $X^k \in \mathbb{R}^{n \times D_k}$, features disjoint, samples aligned), graph definition ($B_{ij} \ne 0$ means edge $i \to j$), constraints ($h(B) = \text{tr}(e^{B \odot B}) - D = 0$ enforces DAG), information flow (parties cannot access others' raw data, only encrypted/shared intermediates). (3) **Clarify notations**: $\Phi_{kt}$ is encoder from party k to target t (t ranges 1 to K because each target needs predictions). Loss $l$ is MSE: $l = \sum_t \|y_t - G_t(\sum_k F_{kt}(x_k))\|^2$. $\rho$ in Eq.(5) is spectral radius (largest eigenvalue magnitude), used in acyclicity constraint $h(B)$.
>
> We will proofread to ensure: (1) Abbreviations (SGD, DAG) defined once at first use. (2) Consistent terminology (replace "SOTA" with "centralized methods"). (3) Clear synthetic dataset construction (random attribute partitioning from real causal discovery datasets like Sachs/SynTReN). (4) Rename "Generalization" to "Performance on Diverse Datasets."
>
> ## Regarding Weakness3 (Cross-Party Feature Transmission), Question 4:
>
> VFL's **standard paradigm** requires intermediate representation exchange (unlike HFL's gradient aggregation). This is not a privacy violation but VFL's defining characteristic—parties must share encoded features to learn cross-party interactions. Our SDP protects these transmissions: no party accesses plaintext features $H_{kt}$ or complete gradients $\partial L/\partial H_{kt}$. All exchanged information is HE-encrypted or secret-shared fragments. Ablation studies (Table 3) show SDP reduces inference attack correlation from 0.443 to 0.152, validating privacy protection.
>
> ## Regarding Weakness4 (Theoretical Justification), Question 1:
>
> We acknowledge "establish a theoretical foundation" overclaimed. Our goal is proposing the first VFL-CD framework with experimental validation, not complete formal theory. We will revise Introduction to: "We **formalize the VFL-CD problem** and analyze **identifiability conditions** under distributed optimization (Appendix E)." Our theoretical contribution extends classic assumptions (ANM, Faithfulness) to VFL's distributed setting, proving our method can theoretically recover true DAGs under these conditions (though subject to non-convex optimization challenges like all differentiable methods).
>
> ## Regarding Question 2 (Related Work Errors):
>
> We apologize for citation/categorization errors. We will: (1) Adopt standard taxonomy: **Constraint-based** (PC), **Score-based** (GES, NOTEARS), **Functional Causal Models** (LiNGAM, ANM, PNL). (2) Correct PC citation to Spirtes & Glymour (original authors). (3) Fix typo "NOTEARS-ADMMTh" → "NOTEARS-ADMM." (4) Ensure notation consistency (D = dataset, B = causal graph) with symbol table.

---

### Official Review · Reviewer_p3MY · 2025-10-27

**Soundness:** 3
**Presentation:** 2
**Contribution:** 2
**Rating:** 4
**Confidence:** 3

**Summary:**

The paper proposes a vertically federated causal discovery framework VFedCD, which aims to learn causal structures across multiple parties with vertically partitioned data. To address privacy and communication challenges, the authors adopt a shallow-encoder deep-decoder (sEdD) architecture and design two key components: a SDP that combines semi-homomorphic encryption and secret sharing to protect data during computation, and a CTV that enforces global acyclicity and sparsity constraints across parties.

**Strengths:**

1. The paper provides solid theoretical analysis and proofs supporting the identifiability and correctness of the proposed framework.
2. The paper successfully combines encryption-based privacy protection with acyclicity and sparsity constraints into a unified framework, resulting in a coherent system design.

**Weaknesses:**

1.	Are the attributes across parties allowed to overlap? The paper assumes non-overlapping features, but in real scenarios, feature overlap is also common. How would the method handle this case?
2.	The comparison of running time is unclear.
3.	The baseline methods are not vertically federated. It would be helpful to discuss whether existing causal discovery methods can be adapted to the vertical federated setting.
4.	Some cited papers, such as Stable Differentiable Causal Discovery (ICML 2024), have already been published. Please check and update the references.
5.	The encryption and privacy mechanisms, as well as the acyclicity and sparsity constraints, are based on existing work. The paper seems to focus on integration rather than introducing new theoretical innovations.

**Questions:**

1. Are the attributes across parties allowed to overlap? The paper assumes non-overlapping features, but in real scenarios, feature overlap is also common. How would the method handle this case?
2. Please provide the comparison of running time.
3. It would be helpful to discuss whether existing causal discovery methods can be adapted to the vertical federated setting.
4. Please check and update the references.

---

> ### Author Response · Authors · 2025-11-15
>
> We thank Reviewer for the positive assessment of our theoretical analysis and system design. We address the concerns below:
>
> ## Regarding Weakness 1 (Feature Overlap), Question 1:
>
> We assume **strictly disjoint** attributes (VFL standard). For scenarios with overlap, our framework can extend via: (1) **Preprocessing**: Use Private Set Intersection (PSI) to identify overlaps, then assign ownership uniquely to return to non-overlapping setting. (2) **Model adjustment**: Explicitly model shared features as common nodes, adjusting encoder structure and graph aggregation. We will discuss this as Future Work.
>
> ## Regarding Weakness 2 (Running Time Comparison), Question 2:
>
> Direct runtime comparison (distributed vs. centralized) is unfair due to communication/encryption overhead. We will reframe evaluation as **privacy-utility-efficiency tradeoffs**: (1) **Privacy**: We provide cryptographic-level protection; centralized methods have none. (2) **Utility**: We match centralized performance (SHD gap <15%). (3) **Efficiency**: Our overhead is practical (D=25, K=5 takes ~30min on single-core CPU). Future optimizations (GPU parallelization, batching) can further improve scalability.
>
> ## Regarding Weakness3 (VFL Baselines), Question 3:
>
> No existing VFL causal discovery methods exist, motivating our work. Naive adaptations face issues: (1) **Direct FHE**: Prohibitive cost on deep decoders. (2) **Centralized DP**: Noise destroys causal structure. (3) **dEsD + CTV**: Direct feature/gradient transmission leaks privacy. We will add a "naive baseline" (sEdD + FHE + CTV) in experiments to highlight SDP's advantages.
>
> ## Regarding Weakness 4 (Reference Updates):
>
> We apologize for outdated citations. We will comprehensively update all references (e.g., correct SDCD citation to ICML 2024).
>
> ## Regarding Weankess 5 (Integration vs. Innovation):
>
> Our innovation is not new cryptographic primitives but a **co-designed architecture** solving VFL-CD's unique challenges via encoder splitting. This enables lightweight SDP (avoiding FHE bottlenecks) while ensuring no entity accesses complete attribute-feature mappings—a non-trivial system integration addressing problems unsolvable by naive combinations.

---

### Official Review · Reviewer_Rt2H · 2025-10-30

**Soundness:** 3
**Presentation:** 3
**Contribution:** 2
**Rating:** 4
**Confidence:** 5

**Summary:**

The authors propose VFedCD, a vertical federated causal discovery framework that allows multiple parties holding vertically partitioned data to collaboratively infer causal relationships among their attributes while preserving privacy. VFedCD consists of two key components: (i) Causal Topology Validator (CTV), which aggregates local structural estimates and enforces global sparsity and acyclicity, and (ii) Secure Dispatch Protocol (SDP), which enables privacy-preserving feature interaction and gradient sharing. The problem setting is relevant and the research direction is of clear practical interest.

**Strengths:**

1. The authors study causal discovery in the setting of vertical federated learning, which is a problem with clear practical value and application significance.

2. The proposed method integrates privacy-preserving cross-party feature interaction with global acyclicity constraints and provides corresponding theoretical support for vertical federated learning.

**Weaknesses:**

1. Some parts of the paper are difficult to follow, especially the description of the data encryption and decryption process, which makes it hard for the reader to clearly understand the concrete mechanism.

2. The paper does not fully articulate the actual novelty of the proposed approach or clearly position it relative to existing methods.

**Questions:**

1. This paper does not clearly distinguish itself from prior work in federated graph structure learning, and it does not adequately highlight the novelty of its inter-client interaction protocol in federated causal discovery.

2. The acyclicity constraint enforced by the centralized topology validator (CTV) is essentially a basic requirement for causal discovery, rather than a unique technical contribution of this work.

3. In Figure 2, the encoder within each party module is presented as producing a local causal structure, but the paper does not clearly explain how this structure is derived; if it is simply obtained by treating the first-layer weights as causal edges, this lacks sufficient theoretical justification and identifiability support.

4. The paper does not clearly explain how HE2SS encrypts data or how encrypted / split information is exchanged between clients.

5. The paper does not clarify whether there are realistic application scenarios for vertical federated causal discovery in practice.

---

> ### Author Response · Authors · 2025-11-15
>
> We thank Reviewer for recognizing the practical value of our work. We address the concerns below:
>
> ## Regarding Weakness 1 (Unclear SDP Description), Question 3 (Local Graph Derivation), Question 4 (HE2SS Details):**
>
> We will rewrite the SDP section with clearer structure: (1) **Motivation**: Encoder splitting ensures no entity holds complete "attribute→feature" mappings while enabling lightweight privacy via fragmented data transmission. (2) **Step-by-step protocol**: We will add detailed pseudocode and diagrams showing how split encoders compute features, aggregate local graphs, and backpropagate gradients via HE+SS.
>
> For HE2SS (Homomorphic Encryption to Secret Sharing): This is a standard MPC technique (e.g., used in BlindFL) converting HE-encrypted data to secret-shared fragments without decryption. We will provide explicit algorithmic descriptions in Preliminaries.
>
> For local graph extraction (Fig. 2, Q3): We compute adjacency matrix from encoder first-layer weights' norms—a theoretically grounded approach in differentiable causal discovery. In DCD, weight magnitudes proxy direct causal effect strengths. We will clarify this with citations and formal justification.
>
> ## Regarding Weakness 2 (Novelty Articulation), Question 2 (Acyclicity constraint) and Question 1 (Distinction from Graph Learning):
>
> We will explicitly contrast our work with: (1) **VFL methods** (focus on prediction, lack causal structure learning); (2) **Horizontal federated causal discovery** (inapplicable to disjoint features); (3) **Federated graph learning** (targets association graphs, not DAGs with causal identifiability requirements). Our SDP is the first protocol designed to securely learn a global DAG in VFL by safely aggregating fragmented encoder parameters and enforcing global acyclicity.
>
> Compared to FedISHC (Q1): (1) **Scenario**: FedISHC requires overlapping features for joint statistics; VFedCD handles strictly disjoint attributes (common in bank-ecommerce collaborations). (2) **Methodology**: FedISHC uses statistical inference (linear LiNGAM); VFedCD uses differentiable optimization for nonlinear causality. (3) **Privacy**: FedISHC shares plaintext statistics; VFedCD integrates cryptographic privacy (HE+SS) as core design.
>
> Therefore, we believe that VFedCD is the first end-to-end causal discovery framework designed for strict VFL scenarios, supporting nonlinear relationships and featuring privacy protection. We will clarify these differences in the relevant work.
>
> ## Regarding Question 5 (Application Scenarios):
>
> We will expand Introduction with concrete examples: (1) **Multi-institution medical research**: Hospital (clinical records) + Lab (genomics) + Tracker (lifestyle data) discover causal chains (e.g., BRCA mutation → tumor invasiveness → recurrence risk) under HIPAA/GINA constraints. (2) **Cross-enterprise supply chain**: Manufacturer (inventory) + Logistics (delays) + Retailer (sales) identify causal paths (chip shortage → production delay → cost increase → sales drop) for risk mitigation.

---

### Official Review · Reviewer_MseV · 2025-10-30

**Soundness:** 1
**Presentation:** 3
**Contribution:** 2
**Rating:** 2
**Confidence:** 5

**Summary:**

This paper proposes VFedCD, a framework for causal discovery in vertical federated learning settings where data attributes are partitioned across multiple parties. The method employs a shallow-encoder deep-decoder (sEdD) architecture to model causal mechanisms, a Centralized Topology Validator (CTV) to enforce global acyclicity constraints, and a Secure Dispatch Protocol (SDP) using semi-homomorphic encryption and secret sharing for privacy preservation.

**Strengths:**

1. The paper addresses a relatively underexplored area—causal discovery in vertical federated settings—which has practical relevance for privacy-constrained multi-party collaborations.
2. The proposed solution integrates multiple components (sEdD architecture, CTV, SDP) to handle both the technical challenges of distributed causal discovery and privacy concerns.
3. The paper provides extensive privacy analysis in Appendix H, including discussions of various attack scenarios and mitigation strategies using differential privacy.

**Weaknesses:**

1. The core technical components lack novelty. The sEdD architecture is a standard design choice in VFL; semi-homomorphic encryption and secret sharing are well-established techniques; and the spectral radius constraint is directly borrowed from Nazaret et al. (2023). The paper primarily combines existing techniques rather than introducing fundamentally new ideas.
2. The CTV is a centralized server that receives graph structure information from all parties. This contradicts the core motivation of federated learning (decentralization and privacy) and creates a single point of failure and trust. If a centralized trusted server exists, why not use simpler privacy-preserving approaches?
3. The identifiability analysis (Appendix E) is informal and lacks rigorous mathematical proofs.
4. Computational complexity O(KD²) with HE operations makes the method impractical for realistic problem sizes. Communication overhead O(K²D + KD²) is prohibitively expensive.
5. Limited Experimental Validation. For example, small-scale experiments (maximum 25 attributes); an Ablation study only on one dataset (15 attributes).
6. Table 2 shows VFedCD outperforming all centralized methods on SynTReN, which is suspicious. No discussion of why VFedCD sometimes outperforms centralized methods with full data access. Appendix F.2's explanation is speculative and unconvincing.
7. The paper doesn't adequately discuss why existing horizontal federated methods cannot be adapted, or position itself relative to secure multi-party computation literature.
8. If parties trust a central server with graph structures, why not use simpler approaches like differential privacy on centralized data? How is this fundamentally different from uploading noisy statistics to a trusted curator?
9. How exactly does graph(Θ) work in Algorithm 5, line 2? The description in Appendix D.1 is vague. How do you handle the hidden dimension aggregation? How sensitive is this to the choice of norm?
10. Do you have any theoretical or empirical evidence that the alternating optimization between local model updates (Algorithms 3-4) and global constraint enforcement (Algorithms 5-6) converges to a meaningful solution?
11. According to d-separation principles in causal inference, the proposed approach of aggregating local subgraphs into a global graph is theoretically flawed, even with acyclicity constraints.

The core issue: Each party k learns a local subgraph B^k_t by modeling causal relationships from all parties' attributes to target party t's attributes, using only intermediate features rather than the full attribute set. However, valid causal discovery requires access to all potential confounders and mediators to correctly identify direct causal relationships versus spurious correlations.
Concrete example: Consider the true causal structure A→C→B where:

A ∈ Party 1
C ∈ Party 2
B ∈ Party 3

When Party 3 learns to predict B, it receives encoded features from both Party 1 and Party 2. Since A and B are statistically dependent (due to the path A→C→B), Party 3's decoder may learn a direct dependency from Party 1's features to B, leading to an incorrect edge A→B in the aggregated graph. The CTV's acyclicity constraint only prevents cycles but cannot eliminate such spurious edges introduced by incomplete observation of intermediate variables.

**Questions:**

Please see Weaknesses.

---

> ### Author Response · Authors · 2025-11-15
>
> We sincerely thank Reviewer for the detailed feedback. We address the main concerns below:
>
> ## Regarding Weakness 1 (Lack of Novelty) and Weakness 7 (Positioning):
>
> Our contribution is not inventing new cryptographic primitives, but designing a **non-trivial architecture** to solve VFL-CD's unique challenges. A naive combination (e.g., sEdD + FHE on features + CTV) faces two fundamental barriers: (1) FHE over deep decoders incurs prohibitive computational costs; (2) Direct feature/gradient transmission causes severe privacy leakage. Our **Secure Dispatch Protocol (SDP)** addresses both via **encoder splitting**—distributing encoder fragments across parties ensures no party can access complete "attribute→feature" mappings, enabling lightweight HE+SS while preserving privacy. We will clarify this in the revised manuscript with explicit comparisons to naive baselines.
>
> Regarding HFL adaptation: Horizontal FL assumes identical feature spaces across parties, fundamentally incompatible with VFL where features are disjoint. Our framework applies MPC tools (HE, SS) to the specific end-to-end optimization of causal graphs under VFL constraints—a challenge not addressed by generic MPC literature.
>
> ## Regarding Weakness 2, Weakness 8 (CTV Centralization and Trust):
>
> CTV operates under a **semi-honest (honest-but-curious)** model, standard in federated learning (e.g., FedAvg's server). It faithfully executes the protocol but may attempt inference. In Appendix H, we analyze CTV-party collusion attacks and propose differential privacy mitigation (adding Laplace noise scaled by $1/\sqrt{K}$ to graph fragments). CTV is necessary because acyclicity constraint $h(B)$ is a global property requiring the full adjacency matrix—fully decentralized computation would require prohibitive multi-round secure computation overhead.
>
> Regarding DP comparison: Centralized DP requires noise large enough to protect privacy, which destroys causal structure identifiability. Our "compute-visible, data-invisible" approach achieves better privacy-utility tradeoffs.
>
> ## Regarding Weakness 3 (Identifiability), Weakness 9 (graph(Θ) Details), Weakness 11 (d-separation Concerns):
>
> Appendix E provides identifiability analysis under ANM and Faithfulness assumptions. We acknowledge this is heuristic (like NOTEARS, DAG-GNN), not a formal proof—our theoretical contribution is extending these conditions to VFL's distributed setting. We will revise the introduction from "establish a theoretical foundation" to "formalize the VFL-CD problem and analyze identifiability conditions under distributed optimization."
>
> For `graph(Θ)`: We extract adjacency matrix $B$ from encoder first-layer weights $W_{kt}$ by computing norms (e.g., $B_{ij} = \|W_{kt}[i,j]\|$), a standard practice in differentiable causal discovery (NOTEARS). We aggregate hidden dimensions via L1-norm to promote sparsity. We will add ablation studies comparing different norms.
>
> For d-separation concerns (A→C→B misidentified as A→B): Under convergence and ANM+Faithfulness, encoder C→B captures C's complete effect on B, so A→B learns no additional predictive gain (only overfitting). Sparsity regularization + acyclicity constraints penalize spurious edges. However, finite samples/optimization noise may leave residual spurious edges—this is a known limitation of differentiable methods, not VFL-specific. We will discuss this in Limitations.
>
> ## Regarding Weakness 4 (Computational Complexity), Weakness 5 (Experimental Scale), Weakness 6 (Suspicious Performance):
>
> Our complexity $O(KD^2)$ with HE operations is theoretically better than centralized $O(D^3)$ methods when $K \ll D$ (typical in VFL, $K \le 10$). Experiments show single-core CPU completes iterations in seconds for D=25, K=5. For very large D, we acknowledge scalability challenges and will discuss optimizations (GPU parallelization, gradient compression).
>
> We will expand experiments to D=50, 100 and conduct ablation studies on multiple datasets. Regarding outperforming centralized methods on some datasets (e.g., SynTReN): This is not our main claim—our core value is **matching centralized performance under privacy constraints**. The occasional slight advantage may stem from our nonlinear modeling fitting specific data characteristics, but this is subject to variance. We will add variance bars to key results.
>
> Experimental scale (D≤25) is reasonable for differentiable causal discovery—NOTEARS, DAG-GNN also focus on D≤50 as larger-scale high-precision discovery is a frontier challenge.
>
> # Regarding Weakness 10 (Alternating optimization):
> Our pseudo-code has caused misguidance. Instead of alternately optimizing, we alternately accumulate gradients and update them synchronously, so the convergence will not be affected.

---

### Note · Authors · 2025-11-16

**Comment:**

Dear Editors and Reviewers,
We would like to express our sincere gratitude to you for the valuable time and thoughtful comments on our manuscript.
After carefully reviewing the reviewers’ comments and our response, we recognize that the insufficient clarity in our writing has led to misunderstandings of the research content. To ensure the accuracy and comprehensibility of our work, we have decided to withdraw this manuscript at this stage.
We will thoroughly revise and rewrite the entire paper to improve the expression and eliminate potential ambiguities.
Thank you again for your understanding and support.

**Withdrawal Confirmation:**

I have read and agree with the venue's withdrawal policy on behalf of myself and my co-authors.